# Graph diffusion distance: Properties and efficient computation

**C. B. Scott**[ID]*, **Eric Mjolsness**

Department of Computer Science, University of California, Irvine, CA, United States of America

* scottcb@uci.edu

**Data Availability Statement:** All relevant data are within the paper and its Supporting information files.

**Funding:** This study was funded in the form of grants by Human Frontiers Science Program [grant HFSP - RGP0023/2018] awarded to EM (hsfp.org);

## Abstract

We define a new family of similarity and distance measures on graphs, and explore their theoretical properties in comparison to conventional distance metrics. These measures are defined by the solution(s) to an optimization problem which attempts find a map minimizing the discrepancy between two graph Laplacian exponential matrices, under norm-preserving and sparsity constraints. Variants of the distance metric are introduced to consider such optimized maps under sparsity constraints as well as fixed time-scaling between the two Laplacians. The objective function of this optimization is multimodal and has discontinuous slope, and is hence difficult for univariate optimizers to solve. We demonstrate a novel procedure for efficiently calculating these optima for two of our distance measure variants. We present numerical experiments demonstrating that (a) upper bounds of our distance metrics can be used to distinguish between lineages of related graphs; (b) our procedure is faster at finding the required optima, by as much as a factor of $10^3$; and (c) the upper bounds satisfy the triangle inequality exactly under some assumptions and approximately under others. We also derive an upper bound for the distance between two graph products, in terms of the distance between the two pairs of factors. Additionally, we present several possible applications, including the construction of infinite "graph limits" by means of Cauchy sequences of graphs related to one another by our distance measure.

## 1 Introduction

Structure comparison, as well as structure summarization, is a ubiquitous problem, appearing across multiple scientific disciplines. In particular, many scientific problems (e.g. inference of molecular properties from structure, pattern matching in data point clouds and scientific images) may be reduced to the problem of inexact graph matching: given two graphs, compute a measure of similarity that gainfully captures structural correspondence between the two. Similarly, many algorithms for addressing multiple scales of dynamical behavior rely on methods for automatically coarsening some model architecture.

In this work we present a graph distance metric, based on the Laplacian exponential kernel of a graph. This measure generalizes the work of Hammond et al. [1] on graph diffusion distance for graphs of equal size; crucially, our distance measure allows for graphs of inequal size. We formulate the distance measure as the solution to an optimization problem dependent on

National Institute of Health [grant R01 HD073179]
awarded to EM (NIH.gov); National Institute of
Aging [Grant R56AG059602] awarded to EM (nia.
nih.gov); National Science Foundation NRT Award
number 1633631 awarded to CBS (nsf.gov);
United States Air Force under Contract No.
FA8750-14-C-0011 under the DARPA PPAML
program awarded to EM (darpa.mil); and the
University of California, Irvine [ICS Research Award
- Endeavor April 2020] awarded to EM (https://
www.ics.uci.edu).

**Competing interests:** The authors have declared
that no competing interests exist.

a comparison of the two graph Laplacians. This problem is a nested optimization problem
with the innermost layer consisting of multivariate optimization subject to matrix constraints
(e.g. orthogonality). To compute this dissimilarity score efficiently, we also develop and dem-
onstrate the lower computational cost of an algorithm which calculates upper bounds on the
distance. This algorithm produces a prolongation/restriction operator, $P$, which produces an
optimally coarsened version of the Laplacian matrix of a graph. Prolongation/restriction oper-
ators produced via the method in this paper have previously been applied to accelerate the
training of machine learning algorithms in [2].

## 1.1 Prior work

Quantitative measures of similarity or dissimilarity between graphs have been studied for
decades owing to their relevance for problems in pattern recognition including structure-
based recognition of extended and compound objects in computer vision, prediction of chemi-
cal similarity based on shared molecular structure, and many other domains. Related problems
arise in quantitative modeling, for example in meshed discretizations of partial differential
equations and more recently in trainable statistical models of data that feature graph-like mod-
els of connectivity such as Bayes Networks, Markov Random Fields, and artificial neural net-
works. A core problem is to define and compute how "similar" two graphs are in a way that is
invariant to a permutation of the the vertices of either graph, so that the answer doesn't depend
on an arbitrary numbering of the vertices. On the other hand unlike an arbitrary numbering,
problem-derived semantic *labels* on graph vertices may express real aspects of a problem
domain and may be fair game for detecting graph similarity. The most difficult case occurs
when such labels are absent, for example in an unstructured mesh, as we shall assume. Here
we detail several measures of graph dissimilarity, chosen by historical significance and similar-
ity to our measure.

   We mention just a few prior works to give an overview of the development of graph dis-
tance measures over time, paying special attention to those which share theoretical or algorith-
mic characteristics with the measure we introduce. Our mathematical distinctions concern the
following properties:

- Does the distance measure require an inner optimization loop? If so is it mainly a discrete or
  continuous optimization formulation?

- Does the distance measure calculation naturally yield some kind of explicit *map* from real-
  valued functions on vertices of one graph to functions on vertices of the other? (A map from
  vertices to vertices would be a special case.) If we use the term "graph signal" to mean a func-
  tion $f: V(G_1) \rightarrow S$ which identifies each vertex of a graph $G_1$ with some state $s \in S$, then a map-
  explicit graph distance is one whose calculation yields a second function $g: V(G_2) \rightarrow V(G_1)$,
  with the composite function $f \circ g: V(G_2) \rightarrow S$.

- Is the distance metric definable on the spectrum of the graph alone, without regard to other
  data from the same graph? The "spectrum" of a graph is a graph invariant calculated as the
  eigenvalues of a matrix related to the adjacency matrix of the graph. Depending on context,
  the spectrum can refer to eigenvalues of the adjacency matrix, graph Laplacian, or normal-
  ized graph Laplacian of a graph. We will usually take the underlying matrix to be the graph
  Laplacian, defined in detail in Section 1.3. Alternatively, does it take into account more
  detailed "structural" aspects of the graph? This categorization (structural vs. spectral) is simi-
  lar to that introduced in [3].

For each of the graph distance variants discussed here, we label them according to the above taxonomy. For example, the two prior works by Eschera et. al. and Hammond et al (discussed in Sections 1.1.4 and 1.1.5) would be labelled as (structural, explicit, disc-opt) and (spectral, implicit, non-opt), respectively. Our distance measure would be labelled (spectral, explicit, cont-opt).

**1.1.1 Quadratic matching of points and graphs (structural, explicit, cont-opt).** Other work focuses on the construction of a point-to-point correspondence between the vertices of two graphs. Gold et. al. [4] define the dissimilarity between two unlabelled weighted graphs (with adjacency matrices $A^{(1)}$ and $A^{(2)}$ and $n_1$ and $n_2$ vertices, respectively) as the solution to the following optimization problem (for real-valued $M = [m_{ij}]$:

$$\text{minimize} \quad \sum_{j=1}^{n_2} \sum_{k=1}^{n_1} \left( \sum_{l=1}^{n_2} A_{jl}^{(1)} m_{lk} - \sum_{p=1}^{n_1} m_{jp} A_{pk}^{(2)} \right)^2 \quad = \left\| A^{(1)} M - M A^{(2)} \right\|_F^2$$

$$\text{subject to} \quad \sum_{i=1}^{n_2} m_{ij} = 1, \qquad\qquad j = 1 \ldots n_1$$

$$\sum_{j=1}^{n_1} m_{ij} = 1, \qquad\qquad i = 1 \ldots n_2$$

$$m_{ij} \geq 0 \qquad\qquad i = 1 \ldots n_2$$
$$j = 1 \ldots n_1$$

(1)

where $\|\cdot\|_F^2$ is the squared Frobenius norm. This problem is similar in structure to the optimization considered in Section 4: a key difference being that Gold et al. consider optimization over real-valued matchings between graph vertices, whereas we consider 0-1 valued matchings between the eigenvalues of the graph Laplacians. In [5, 6] the authors present computational methods for computing this optimum, and demonstrate applications of this distance measure to various machine learning tasks such as 2D and 3D point matching, as well as graph clustering. Gold et al. also introduce the *softassign*, a method for performing combinatorial optimization with both row and column constraints, similar to those we consider.

**1.1.2 Cut-distance of graphs (structural, implicit, disc-opt).** Lovász [7] defines the *cut-distance* of a pair of graphs as follows: Let the □-norm of a matrix $B$ be given by:

$$\|B\|_\square = \frac{1}{n^2} \max_{S,T \subseteq 1\ldots n} \left| \sum_{i \in S, j \in T} B_{ij} \right|$$

(2)

Given two labelled graphs $G_1, G_2$, on the same set of vertices, and their adjacency matrices $A_1$ and $A_2$, the cut-distance $d_{\text{cut}}(G_1, G_2)$ is then given by

$$D_{\text{cut}}(G_1, G_2) = \|A_1 - A_2\|_\square$$

(3)

(for more details, see [7]). Computing this distance requires combinatorial optimization (over all vertex subsets of $G_1, G_2$) but this optimization does not result in an explicit map between $G_1$ and $G_2$.

**1.1.3 Wasserstein earth mover distance (spectral, implicit, disc-opt).** One common metric between graph spectra is the Wasserstein Earth Mover Distance. Most generally, this distance measures the cost of transforming one probability density function into another by moving mass under the curve. If we consider the eigenvalues of a (possibly weighted) graph as point masses, then the EMD measures the distance between the two spectra as the solution to a

transport problem (transporting one set of points to the other, subject to constraints e.g. a limit on total distance travelled or a limit on the number of 'agents' moving points). The EMD has been used in the past in various graph clustering and pattern recognition contexts; see [8]. In the above categorization, this is an optimization-based spectral distance measure, but is implicit, since it does not produce a map from vertices of $G_1$ to those of $G_2$ (informally, this is because the EMD is not translating one set of eigenvalues into the other, but instead transforming their respective histograms). Recent work applying the EMD to graph classification includes [9, 10]. Some similar recent works [11, 12] have used optimal transport theory to compare graphs. In this framework, signals on each graph are smoothed, and considered as draws from probability distribution(s) over the set of all graph signals. An optimal transport algorithm is used to find the optimal mapping between the two probability distributions, thereby comparing the two underlying graphs.

**1.1.4 Graph-edit distance.** The graph edit distance measures the total cost of converting one graph into another with a sequence of local edit moves, with each type of move (vertex deletion or addition, edge deletion or addition, edge division or contraction) incurring a specified cost. Costs are chosen to suit the graph analysis problem at hand; determining a cost assignment which makes the edit distance most instructive for a certain set of graphs is an active area of research. The distance measure is then the sum of these costs over an optimal sequence of edits, which must be found using some optimization algorithm i.e. a shortest-path algorithm (the best choice of algorithm may vary, depending on how the costs are chosen). The sequence of edits may or may not (depending on the exact set of allowable edit moves) be adaptable into an explicit map between vertex-sets. Classic pattern recognition literature includes: [13–16].

**1.1.5 Diffusion distance due to Hammond et al. [1].** We discuss this recent distance metric more thoroughly below. This distance measures the difference between two graphs as the maximum discrepancy between probability distributions which represent single-particle diffusion beginning from each of the nodes of $G_1$ and $G_2$. This distance is computed by comparing the eigenvalues of the heat kernels of the two graphs. The optimization involved in calculating this distance is a simple unimodal optimization over a single scalar, $t$, representing the passage of time for the diffusion process on the two graphs; hence we do not count this among the "optimization based" methods we consider.

**1.1.6 Novel diffusion-derived measures.** In this work, we introduce a family of related graph distance measures which compare two graphs in terms of similarity of a set of probability distributions describing single-particle diffusion on each graph. For two graphs $G_1$ and $G_2$ with respective Laplacians $L(G_1)$ and $L(G_2)$, the matrices $e^{tL(G_1)}$ and $e^{tL(G_2)}$ are called the *Laplacian Exponential Kernels* of $G_1$ and $G_2$ ($t$ is a scalar representing the passage of time). The column vectors of these matrices describe the probability distribution of a single-particle diffusion process starting from each vertex, after $t$ time has passed. The norm of the difference of these two kernels thus describes how different these two graphs are, from the perspective of single-particle diffusion, at time $t$. Since these distributions are identical at very-early and very-late times $t$ (we formalize this notion in Section 2.1), a natural way to define a graph distance is to take the supremum over all $t$. When the two graphs are the same size, so are the two kernels, which may therefore be directly compared with a matrix norm. This case is the case considered by Hammond et al. [1]. However, to compare two graphs of different sizes, we need a mapping between the column vectors of $e^{tL(G_1)}$ and $e^{tL(G_2)}$.

Optimization over a suitably constrained prolongation/restriction operator between the graph Laplacians of the two graphs is a permutation-invariant way to compare the behavior of a diffusion process on each. The prolongation map $P$ thus calculated may then be used to map signals (by which we mean values associated with vertices or edges of a graph) on $G_1$ to the

space of signals on $G_2$ (and vice versa). In [2] we implicitly consider the weights of an artificial neural network model to be graph signals, and use these operators to train a hierarchy of linked neural network models. However, in that work we do not address efficient calculation of this distance or the associated operators, a major focus of this paper.

We also, in sections 3.2 and 3.3 consider a time conversion factor between diffusion on graphs of unequal size, and consider the effect of limiting this optimization to sparse maps between the two graphs (again, our case reduces to Hammond when the graphs in question are the same size, dense $P$ and $R$ matrices are allowed, and our time-scaling parameter is set to 1).

In this work, we present an algorithm for computing the type of nested optimization given in our definition of distance (Eqs 8 and 9). The innermost loop of our distance measure optimization consists of a Linear Assignment Problem (LAP, defined below) where the entries of the cost matrix have a nonlinear dependence on some external variable. Our algorithm greatly reduces both the count and size of calls to the external LAP solver. We use this algorithm to compute an upper bound on our distance measure, but it could also be useful in other similar nested optimization contexts: specifically, nested optimization where the inner loop consists of a linear assignment problem whose costs depend quadratically on the parameter in the outermost loop.

## 1.2 Background

The ideal for a quantitative measure of similarity or distance on some set $S$ is usually taken to be a distance *metric* $d : S \times S \mapsto \mathbb{R}$ satisfying for all $x, y, z \in S$:

- Non-negativity: $d(x, y) \geq 0$

- Identity: $d(x, y) = 0 \Leftrightarrow x = y$

- Symmetry: $d(x, y) = d(y, x)$

- Triangle inequality: $d(x, z) \leq d(x, y) + d(y, z)$

Then $(S, d)$ is a *metric space*. Euclidean distance on $\mathbb{R}^d$ and geodesic distance on manifolds satisfy these axioms. They can be used to define algorithms that generalize from $\mathbb{R}^d$ to other spaces. A variety of weakenings of these axioms are required in many applications, by dropping some axioms and/or weakening others. For example if $S$ is a set of nonempty sets of a metric space $S_0$, one can define the "Hausdorff distance" on $S$ which is an *extended pseudo-metric* that obeys the triangle inequality but not the Identity axiom and that can take values including $+\infty$. As another example, any measure measure of distance on graphs which is purely spectral (in the taxonomy of Section 1.1) cannot distinguish between graphs which have identical spectra. We discuss this in more detail in Section 2.3.

Additional properties of distance metrics that generalize Euclidean distance may pertain to metric spaces related by Cartesian product, for example, by summing the squares of the distance metrics on the factor spaces. We will consider an analog of this property in Section 3.4.

## 1.3 Definitions

**Graph Laplacian**: For an undirected graph $G$ with adjacency matrix $A$ and vertex degrees $d_1$, $d_2 \ldots d_n$, we define the Laplacian of the graph as

$$\begin{aligned} L(G) \quad &= A - diag(\{d_1, d_2 \ldots d_n\}) \\ &= A - diag(\mathbf{1} \cdot A) \\ &= A(G) - D(G) \end{aligned} \quad (4)$$

$L(G)$ is sometimes instead defined as $D(G) - A(G)$; we take this sign convention for $L(G)$ because it agrees with the standard continuum Laplacian operator, $\Delta$, of a multivariate function $f$: $\Delta f = \sum_{i=1}^{n} \frac{\delta^2 f}{\delta x_i^2}$.

**Frobenius norm**: The squared Frobenius norm, $\|A\|_F^2$ of a matrix $A$ is given by the sum of squares of matrix entries. This can equivalently be written as $\text{Tr}[A^T A]$.

**Linear Assignment Problem (LAP)**: We take the usual definition of the Linear Assignment Problem (see [17, 18]): we have two lists of items $S$ and $R$ (sometimes referred to as "workers" and "jobs"), and a cost function $c : S \times R \rightarrow \mathbb{R}$ which maps pairs of elements from $S$ and $R$ to an associated cost value. This can be written as a linear program for real-valued $x_{ij}$ as follows:

$$
\begin{aligned}
\text{minimize} \quad & \sum_{i=1}^{m} \sum_{j=1}^{n} c(s_i, r_j) x_{ij} \\
\text{subject to} \quad & \sum_{i=1}^{m} x_{ij} \leq 1, \qquad j = 1 \ldots n \\
& \sum_{j=1}^{n} x_{ij} \leq 1, \qquad i = 1 \ldots m \\
& x_{ij} \geq 0 \qquad\qquad i = 1 \ldots m, j = 1 \ldots n
\end{aligned}
\tag{5}
$$

Generally, "Linear Assignment Problem" refers to the square version of the problem where $|S| = |R| = n$, and the objective is to allocate the $n$ jobs to $n$ workers such that each worker has exactly one job and vice versa. The case where there are more workers than jobs, or vice versa, is referred to as a Rectangular LAP or RLAP. In practice, the conceptually simplest method for solving an RLAP is to convert it to a LAP by *augmenting* the cost matrix with several columns (rows) of zeros. In this case, solving the RLAP is equivalent to solving a LAP with size $max(n, m)$. Other computational shortcuts exist; see [19] for details. Since the code we use to solve RLAPs takes the augmented cost matrix approach, we do not consider other methods in this paper.

**Matching**: we refer to a 0-1 matrix $M$ which is the solution of a particular LAP as a "matching". We may refer to the "pairs" or "points" of a matching, by which we mean the pairs of indices $(i, j)$ with $M_{ij} = 1$. We may also say in this case that $M$ "assigns" $i$ to $j$.

**Hierarchical graph sequences**: A Hierarchical Graph Sequence (HGS) is a sequence of graphs, indexed by $l \in \mathbb{N} = 0, 1, 2, 3 \ldots$, satisfying the following:

- $G_0$ is the graph with one vertex and one self-loop, and;

- Successive members of the lineage grow roughly exponentially—that is, there exists some base $b$ such that the growth rate as a function of level number $l$ is $O(b^{l^{1+\epsilon}})$, for all $\epsilon > 0$.

**Graded graph**: A graded graph is a graph along with a vertex labelling, where vertices are labelled with non-negative integers such that $\Delta l$, the difference in label over any edge, is in $\{-1, 0, 1\}$. We will refer to the $\Delta l = 0$ edges as "within-level" and the $l = \pm 1$ edges as "between-level".

**Graph lineages**: A graph lineage is a graded graph with two extra conditions:

- The vertices and edges with $\Delta l = 0$ form a HGS; and

- the vertices and edges with $\Delta l = \pm 1$ form a HGS of bipartite graphs.

More plainly, a graph lineage is an exponentially growing sequence of graphs along with ancestry relationships between nodes. We will also use the term graph lineage to refer to the

HGS in the first part of the definition. Some intuitive examples of graph lineages in this sense are the following:

- Path graphs or cycle graphs of size $b^n$ for any integer $b$.

- More generally, grid graphs of any dimension $d$, of side length $b$, yielding a lineage which grows with size $b^{dn}$ (with periodic or nonperiodic boundary conditions).

- For any probability distribution $p(x, y)$ whose support is points in the unit square, we can construct a graph by discretizing the map of $p$ as a function of $x$ and $y$, and interpreting the resulting matrix as the adjacency matrix of a graph. For a specific probability distribution $p$, the graphs derived this way with discretizations of exponentially increasing bin count form a graph lineage.

- The *triangulated mesh* is a common object in computer graphics [20–22], representing a discretization of a 2-manifold embedded in $R^3$. Finer and finer subdivisions of such a mesh constitute a graph lineage.

Several examples of graph lineages are used in the discussion of the numerical properties of Graph Diffusion Distance in Section 5.1. Additional examples (a path graph and a triangulated mesh) can be found in Figs 1 and 2.

**Box product (□) of graphs**: For two graphs $G$ and $H$ with vertex sets $V(G) = \{g_1, g_2 \ldots g_n\}$ and $V(H) = \{h_1, h_2 \ldots h_m\}$, we say the product graph $G□H$ is the graph with vertex set $V(G□H)$

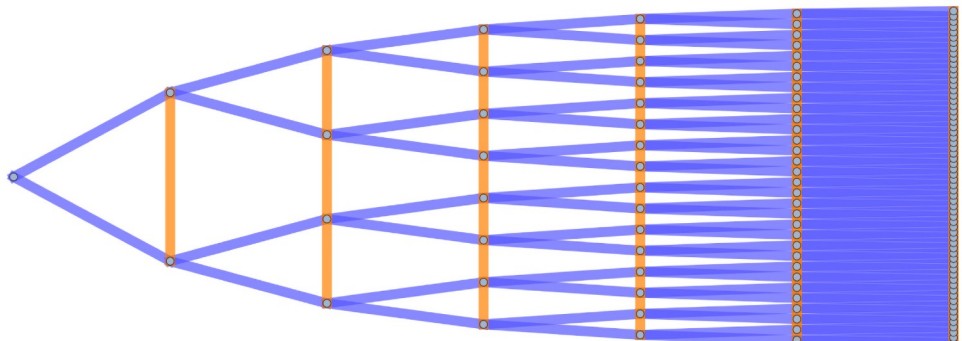

**Fig 1. The first seven levels of the graph lineage of path graphs, with ancestry relationships.** $\Delta l = 0$ edges are colored in orange, $\Delta l = \pm 1$ edges are colored in blue. Self-loops are not illustrated.

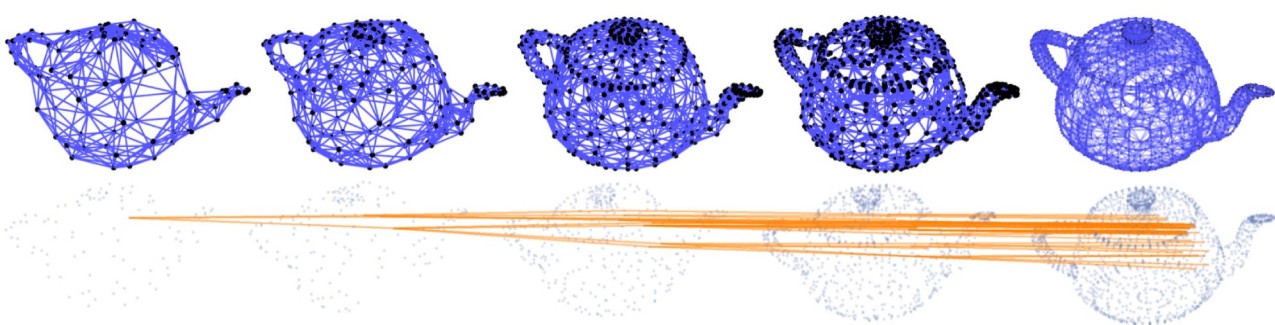

**Fig 2.** Top: subsamples of a mesh of the Utah teapot, of increasing density (each node is connected to its 8 nearest neighbors by the $\Delta l = \pm 0$ edges, rendered in blue). These samples form a graph lineage ($\Delta l = \pm 1$ edges are not illustrated). Bottom: the same set of nodes, with only $\Delta l = \pm 1$ edges plotted (in orange) for one node from the coarsest level and its descendants.

= $V(G) \times V(H)$ and an adjacency relationship defined by: $(g_1, h_1) \sim (g_2, h_2)$ in $G \square H$ if and only if $g_1 \sim g_2$ in $G$ and $h_1 = h_2$, or $g_1 = g_2$ and $h_1 \sim h_2$ in $H$. Note that the adjacency matrix of this relationship may be represented by the following identity:

$$A(G \square H) = A(G) \otimes I_m + I_n \otimes A(H) \tag{6}$$

where $\otimes$ is the Kronecker Product of matrices (See [23], Section 11.4).

## 2 Graph diffusion distance definitions

### 2.1 Diffusion distance definition

We generalize the diffusion distance defined by Hammond et al. [1]. This distortion measure between two graphs $G_1$ and $G_2$, of the same size, was defined as:

$$D_{\text{Hammond}}(G_1, G_2) = \sup_t \lVert e^{tL_1} - e^{tL_2} \rVert_F^2 \tag{7}$$

where $L_i$ represents the graph Laplacian of $G_i$.

This may be interpreted as measuring the maximum divergence, as a function of $t$, between diffusion processes starting from each vertex of each graph, as measured by the squared Euclidean distance between the column vectors of $e^{tL_i}$. Each column $v_j$ of $e^{tL_i}$ (which is called the Laplacian Exponential Kernel) describes a probability distribution of visits (by a random walk of duration $t$, with node transition probabilities given by the columns of $e^L$) to the vertices of $G_i$, starting at vertex $j$. This distance metric is then measuring the difference between the two graphs by comparing these probability distributions; the motivation between taking the supremum over all $t$ is that the value of the objective function at the maximum is the most these two distributions can diverge. See Fig 3 for an example of a distance calculation, with a characteristic peak. For further intuition about why the peak is the most natural place to take as the distance, rather than some other arbitrary time, note that at very early times and very late times, the probability distribution of vertex visits is agnostic to graph structure: at early times no diffusion has had a chance to take place, while at very late times the distribution of

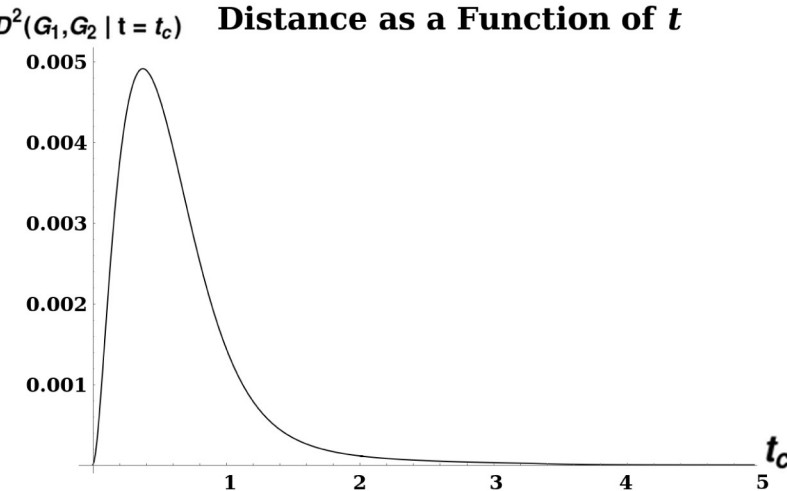

**Fig 3. A plot illustrating unimodality of diffusion distance.** $D^2$ was calculated between two grid graphs $Sq_7$ and $Sq_8$ of size $7 \times 7$ and $8 \times 8$, respectively. The distance is given by the formula $D^2(Sq_7, Sq_8 | t) = \inf_{\alpha > 0} \inf_{P | \mathcal{C}(P)} \lVert P e^{\frac{t}{\alpha}L(Sq_7)} - e^{t\alpha L(Sq_8)} P \rVert_F^2$ as a function of $t$. The peak, at $t \approx .318$, yields the distance $D^2(Sq_7, Sq_8)$.

vertex-visits converges to the stationary state for each connected component of the graph. Hence we are most interested in a regime of $t$-values in between these extremes, where differences in $G_1$ and $G_2$ are apparent in their differing probability distributions.

Our contribution generalizes this measure to allow for graphs of differing size. We add two variables to this optimization: a *prolongation* operator, $P$ (represented as a rectangular matrix), and a time-scaling factor, $\alpha$. The dissimilarity between two graphs $G_1$ and $G_2$ (with Laplacians $L_i = L(G_i)$) is then defined as:

$$D^2(G_1, G_2) \quad = \sup_{t>0} \inf_{\alpha>0} \inf_{P|\mathcal{C}(P)} \left\| Pe^{\frac{t}{\alpha}L_1} - e^{\alpha t L_2}P \right\|_F^2 \tag{8}$$

where $\mathcal{C}(P)$ represents some set of constraints on the matrix $P$. For the remainder of this work we use $D(G_1, G_2)$ to refer to the distance and $D^2(G_1, G_2)$ to refer to the squared distance—this notation is chosen to simplify the exposition of some proofs. It will be convenient for later calculations to introduce and assume the concept of *transitive constraints*—by which we mean that for any constraint $\mathcal{C}$, satisfaction of $\mathcal{C}$ by $P_1$ and $P_2$ implies satisfaction of $\mathcal{C}$ by their product $P_1 P_2$ (when such a product is defined). Some (non-exclusive) examples of transitive constraints include orthogonality, particular forms of sparsity, and their conjunctions.

The simplest transitive constraint we will consider is that $P$ should be orthogonal. Intuitively, an orthogonal $P$ represents a norm-preserving map between nodes of $G_1$ and nodes of $G_2$, so we are measuring how well diffusion on $G_1$ approximates diffusion on $G_2$, as projected by $P$. Note that since in general $P$ is a rectangular matrix it is not necessarily true that $PP^T = I$. We assume that $|G_1| = n_1 \leq n_2 = |G_2|$; if not, the order of the operands is switched, so that $P$ is always at least as wide as it is tall. We also briefly consider a sparsity constraint in section 3.3 below. Since sparsity is more difficult to treat numerically, our default constraint will be orthogonality alone. Other constraints could include bandedness and other structural constraints (see Section 6). We also note that because $L$ is finite-dimensional, the exponential map is continuous and therefore we can swap the order of optimization over $t$ and $\alpha$. The optimization procedure outlined in this paper optimizes these variables in the order presented above (namely: an outermost loop of maximization over $t$, a middle loop of minimization over $\alpha$ given $t$, and an innermost loop of minimization over $P$ given $t$ and $\alpha$).

The other additional parameter, $\alpha$, controls dilation between the passage of time in the two graphs, to account for different scales. Again, the intuition is that we are interested in the difference between structural properties of the graph (from the point of view of single-particle diffusion) independent of the absolute number of nodes in the graph. As an example, diffusion on an $n \times n$ grid is a reasonably accurate approximation of more rapid diffusion on a $2n \times 2n$ grid, especially when $n$ is very large. In our discussion of variants of this dissimilarity score, we will use the notation $D^2(G_1, G_2 | x = c)$ to mean restrictions of any of our distortion measure equations where variable $x$ is held to a constant value $c$; In cases where it is clear from context which variable is held to a fixed value $c$, we will write $D^2(G_1, G_2 | c)$.

At very early times the second and higher-order terms of the Taylor Series expansion of the matrix exponential function vanish, and so $e^{tL} \approx I + tL$. This motivates the *early-time* or "linear" version of this distance, $\tilde{D}$:

$$\tilde{D}^2(G_1, G_2) \quad = \inf_{\alpha>0} \inf_{P|\mathcal{C}(P)} \left\| \frac{1}{\alpha} PL_1 - \alpha L_2 P \right\|_F^2 \tag{9}$$

$$\approx \frac{1}{t^2} \left( \inf_{\alpha>0} \inf_{P|\mathcal{C}(P)} \left\| Pe^{\frac{t}{\alpha}L_1} - e^{\alpha t L_2}P \right\|_F^2 \right) \tag{10}$$

(Note that the identity matrices cancel). The outermost optimization (maximization over $t$) is removed for this version of the distance, as $t$ can be factored out:

$$\left\|\frac{t}{\alpha} PL_1 - \alpha t L_2 P\right\|_F^2 = t^2 \left\|\frac{1}{\alpha} PL_1 - \alpha L_2 P\right\|_F^2 \tag{11}$$

For the exponential version of the dissimilarity score, we note briefly that the supremum over $t$ of our objective function must exist, since for any $G_1, G_2$:

$$D^2(G_1, G_2) \leq D^2\left(G_1, G_2 \middle| \alpha = 1, P = \begin{bmatrix} I \\ 0 \end{bmatrix}\right) \tag{12}$$

In other words, the infimum over all $P$ and $\alpha$ is bounded above by any particular choice of values for these variables. Since

$$D^2 \quad \left(G_1, G_2 \middle| t = 0, \alpha = 1, P = \begin{bmatrix} I \\ 0 \end{bmatrix}\right) = 0, \tag{13}$$

and

$$\lim_{t_c \to \infty} D^2 \quad \left(G_1, G_2 \middle| t_c, \alpha = 1, P = \begin{bmatrix} I \\ 0 \end{bmatrix}\right) = 0 \tag{14}$$

this upper bound must have a supremum (possibly 0) at some $t^* \in [0, \infty)$. Then

$$D^2\left(G_1, G_2 \middle| t^*, \alpha = 1, P = \begin{bmatrix} I \\ 0 \end{bmatrix}\right) \tag{15}$$

must be finite and therefore so must the objective function.

## 2.2 Directedness of distance and constraints

We note that this distance measure, as defined so far, is *directed*: the operands $G_1$ and $G_2$ serve differing roles in the objective function. This additionally makes the constraint predicate $\mathcal{C}(P)$ ambiguous: when we state that $\mathcal{C}$ represents orthogonality, it is not clear whether we are referring to $P^T P = I$ or $PP^T = I$ (only one of which can be true for a non-square matrix $P$). To remove this ambiguity, we will, for the computations in the rest of this manuscript, define the distance metric to be symmetric: the distance between $G_1$ and $G_2$ with $|G_1| \leq |G_2|$ is always $D(G_1, G_2)$. $P$ is then always at least as tall as it is wide, so of the two choices of orthogonality constraint we select $P^T P = I$.

## 2.3 Variants of distance measure

Thus far we have avoided referring to this graph dissimilarity function as a "distance metric". As we shall see later, full optimization of Eqs 8 and 9 over $\alpha$ and $P$ is too loose, in the sense that the distances $D(G_1, G_2)$, $D(G_2, G_3)$, and $D(G_1, G_3)$ do not necessarily satisfy the triangle inequality. The same is true for $\tilde{D}$. See Section 5.3.1 for numerical experiments suggesting a particular parameter regime where the triangle inequality is satisfied. We thus define several restricted/augmented versions of both $D$ and $\tilde{D}$ which are guaranteed to satisfy the triangle inequality. These different versions are summarized in Table 1. These variously satisfy some of the conditions necessary for generalized versions of distance metrics, including:

**Table 1. Summary of this paper's investigation of different forms of our graph dissimilarity measure.** In this work, we systematically explore properties of this measure given sparsity parameter $s = 0$, and various regimes of $t$ (fixed at some early time, or maximized over all $t$) and $\alpha$ (fixed at $\alpha = 1$, fixed at a constant power $r$ of the ratio of graph sizes, or minimized over all $\alpha$. We leave exploration of nonzero values of the sparsity parameter to future work. Variants not explicitly called out are not considered. In the case where $\alpha$ and $t$ are both optimized and $s > 0$, it is unclear which of the metric conditions GDD satisfies, hence the corresponding classification is left blank.

| $t$ | $\alpha$ | $s$ | Classification | Treatment in this manuscript |
|---|---|---|---|---|
| Fixed at $t_c < \epsilon$ | Fixed at $\alpha_c = 1$ | $s = 0$ | Pseudometric | Defined in Eq 18. Optimized by one pass of LAP solver. Triangle inequality proven in Theorem 2. |
| Fixed at $t_c < \epsilon$ | Fixed at $\alpha_c = \left(\frac{n_1}{n_2}\right)^r$ | $s = 0$ | Pseudometric | Defined in Eq (24). Optimized by one pass of LAP solver. Triangle inequality proven in Theorem 4. |
| Fixed at $t_c < \epsilon$ | Optimized | $s = 0$ | Premetric | Defined in Eq 9. Optimized by Algorithm 1. Triangle inequality violations examined experimentally in Section 5.3.1. |
| Optimized | Fixed at $\alpha_c = 1$ | $s = 0$ | Metric | When $|G_1| = |G_2|$, this is Hammond et. al's version of graph distance. |
| Optimized | Optimized | $s = 0$ | Premetric | Defined in Eq 8. Optimized by Algorithm 2. Graph Product upper bound proven in Theorem 5. Triangle inequality violations examined experimentally in Section 5.3.1. Used to calculate graph distances in Sections 5.3.2 and 5.3.3. |
| Fixed at $t_c < \epsilon$ | Fixed at $\alpha_c = 1$ | $s > 0$ | Pseudometric | Triangle inequality proven in Theorem 2. |
| Fixed at $t_c < \epsilon$ | Fixed at $\alpha_c = \left(\frac{n_1}{n_2}\right)^r$ | $s > 0$ | Pseudometric | Triangle inequality proven in Theorem 4. |
| Optimized | Optimized | $s > 0$ | | Discussed in Section 3.3. |

- Premetric: a function $d(x, y)$ for which $d(x, y) \geq 0$ and $d(x, y) = d(y, x)$ for all $x, y$.

- Pseudometric: As a premetric, but additionally $d(x, z) \leq d(x, y) + d(y, z)$ for all $x, y, z$.

- $\rho$-inframetric: As a premetric, but additionally $d(x, z) \leq \rho(d(x, y) + d(y, z))$ and $d(x, y) = 0$ if and only if $x = y$, for all $x, y, z$.

Additionally, we note here that a distance measure on graphs using Laplacian spectra can at best be a pseudometric, since isospectral, non-isomorphic graphs are well-known to exist [24, 25]. Characterizing the conditions under which two graphs are isospectral but not isomorphic is an open problem in spectral graph theory. However, previous computational work has led to the conjecture that "almost all" graphs are uniquely defined by their spectra [26–28], in the sense that the probability of two graphs of size $n$ being isospectral but not isomorphic goes to 0 as $n \to \infty$. Furthermore, our numerical experiments have indicated that the violation of the triangle inequality is bounded, in the sense that $D(G_1, G_3) \leq \rho^*(D(G_1, G_2) + D(G_2, G_3))$ for $\rho \approx$ 2.1. This means that even in the least restricted case our similarity measure may be a 2.1-infra-pseudometric on graphs (and, since such isospectral, non-isomorphic graphs are relatively rare, it behaves like a 2.1-inframetric). As we will see in Section 3, in some more restricted cases we can prove triangle inequalities, making our measure a pseudometric. In Section 4.1, we discuss an algorithm for computing the optima in Eqs (8) and (9). First, we discuss some theoretical properties of this dissimilarity measure.

## 3 Theoretical properties of D(G1, G2)

In this section we prove several properties of various instances of our graph dissimilarity score, including triangle inequalities for some specific versions and an upper bound on the distance between two graph products.

## 3.1 Triangle inequality for $\alpha = 1$

**Lemma 1**. *For any matrices M and P, with P satisfying $P^T P = I$, $\|PM\|_F^2 \leq \|M\|_F^2$ and $\|MP\|_F^2 \leq \|M\|_F^2$.*

*Proof.* Suppose without loss of generality that $P^T P = I$. Then:

1. $\|PM\|_F^2 = \text{Tr}\,[M^T P^T P M] = \text{Tr}\,[M^T M] = \|M\|_F^2$

2. If $P^T P = I$, then letting $PP^T = \Pi$, $\Pi$ is a projection operator satisfying $\Pi^T = \Pi = \Pi^2$. Then,

$$
\begin{aligned}
\|M\|_F^2 = \text{Tr}\,[M^T M] &= \text{Tr}\,[M^T M(\Pi + (I - \Pi))] \\
&= \text{Tr}\,[M^T M\Pi] + \text{Tr}\,[M^T M(I - \Pi)] \\
&= \text{Tr}\,[M^T MPP^T] + \text{Tr}\,[M^T M(I - \Pi)^2] \\
&= \|MP\|_F^2 + \|M(I - \Pi)\|_F^2 \\
&\geq \|MP\|_F^2
\end{aligned}
\tag{16}
$$

**Theorem 2**. *$\tilde{D}^2$ satisfies the triangle inequality for $\alpha = 1$.*

*Proof.* Let $G_1, G_2, G_3$ be simple graphs, with Laplacians $L_1, L_2, L_3$. Let

$$
P_{31} = \arg\inf_{P|\mathcal{C}(P)} \|PL_1 - L_3 P\|_F^2
\tag{17}
$$

Then

$$
\begin{aligned}
\tilde{D}^2(G_1, G_3|\alpha = 1) &= \|P_{31}L_1 - L_3 P_{31}\|_F^2 = \inf_{P|\mathcal{C}(P)} \|PL_1 - L_3 P\|_F^2 \\
&\leq \inf_{P_{32}, P_{21}|\mathcal{C}(P_{32}P_{21})} \|P_{32}P_{21}L_1 - L_3 P_{32}P_{21}\|_F^2
\end{aligned}
\tag{18}
$$

where we write $\mathcal{C}(P_{32}P_{21})$ to signify that the product $P_{32}P_{21}$ satisfies the original transitive constraints on $P$, e.g. orthogonality and/or sparsity. Since the constraint predicate $\mathcal{C}(P)$ satisfies Eq (25), then so does their product, so we may write

$$
\begin{aligned}
\tilde{D}(G_1, G_3|\alpha = 1) &\leq \inf_{P_{32}|\mathcal{C}(P_{32})} \inf_{P_{21}|\mathcal{C}(P_{21})} \|P_{32}P_{21}L_1 - L_3 P_{32}P_{21}\|_F \\
&= \inf_{P_{32}|\mathcal{C}(P_{32})} \inf_{P_{21}|\mathcal{C}(P_{21})} \|P_{32}P_{21}L_1 - P_{32}L_2 P_{21} + P_{32}L_2 P_{21} - L_3 P_{32}P_{21}\|_F \\
&\leq \inf_{P_{32}|\mathcal{C}(P_{32})} \inf_{P_{21}|\mathcal{C}(P_{21})} (\|P_{32}P_{21}L_1 - P_{32}L_2 P_{21}\|_F \\
&\quad + \|P_{32}L_2 P_{21} - L_3 P_{32}P_{21}\|_F) \\
&= \inf_{P_{32}|\mathcal{C}(P_{32})} \inf_{P_{21}|\mathcal{C}(P_{21})} (\|P_{32}(P_{21}L_1 - L_2 P_{21})\|_F \\
&\quad + \|(P_{32}L_2 - L_3 P_{32})P_{21}\|_F)
\end{aligned}
\tag{19}
$$

By Lemma 1,

$$
\begin{aligned}
\tilde{D}(G_1, G_3 | \alpha = 1) \quad &\leq \inf_{P_{32}|\mathcal{C}(P_{32})} \inf_{P_{21}|\mathcal{C}(P_{21})} \big( \|P_{21}L_1 - L_2 P_{21}\|_F \\
&\quad + \quad \|P_{32}L_2 - L_3 P_{32}\|_F \big) \\
&= \inf_{P_{21}|\mathcal{C}(P_{21})} \|P_{21}L_1 - L_2 P_{21}\|_F \\
&\quad + \inf_{P_{32}|\mathcal{C}(P_{32})} \|P_{32}L_2 - L_3 P_{32}\|_F \\
&= \tilde{D}(G_1, G_2 | \alpha = 1) + \tilde{D}(G_2, G_3 | \alpha = 1)
\end{aligned}
\tag{20}
$$

We note that in this proof we use $L_1$, $L_2$, and $L_3$ (making this the small-$t$ or linear version of the objective function), but the same argument holds when all three are replaced with $e^{tL_i}$, so we also have

**Corollary 3**. *D satisfies the triangle inequality for $\alpha = 1$.*

*Proof.* By the same calculation as in Theorem 2, with all $L_i$ replaced by $e^{t_c L_i}$, we have

$$
D(G_1, G_3 | t_c, \alpha = 1) \quad \leq D(G_1, G_2 | t_c, \alpha = 1) + D(G_2, G_3 | t_c, \alpha = 1)
\tag{21}
$$

for any constant $t_c$. Then, letting

$$
t_{13} = \arg\sup_{t_c} D(G_1, G_3 | t_c, \alpha = 1)
\tag{22}
$$

we have:

$$
\begin{aligned}
D(G_1, G_3 | \alpha = 1) \quad &= \sup_{t_c} D(G_1, G_3 | t_c, \alpha = 1) \\
&= D(G_1, G_3 | t_{13}, \alpha = 1) \\
&\leq D(G_1, G_2 | t_{13}, \alpha = 1) + D(G_2, G_3 | t_{13}, \alpha = 1) \\
&\leq \sup_{t_c} D(G_1, G_2 | t_c, \alpha = 1) \\
&\quad + \sup_{t_c} D(G_2, G_3 | t_c, \alpha = 1) \\
&= D(G_1, G_2 | \alpha = 1) + D(G_2, G_3 | \alpha = 1)
\end{aligned}
\tag{23}
$$

Note that in the proofs of Theorem 2, Theorem 4, and Corollary 3, we assume that the constraint predicate $\mathcal{C}(P)$ includes at least orthogonality (so that we may apply Lemma 1). However, this constraint predicate could be more strict, e.g. include both orthogonality and sparsity. Hence these statements also apply to the $s > 0$ cases in Table 1, which we do not otherwise consider in this work: in our numerical experiments we (for reasons of computational simplicity) only require our optimization over $P$ be orthogonally constrained.

## 3.2 Time-Scaled Graph Diffusion Distance

For any graphs $G_1$ and $G_2$, we define the Time-Scaled Graph Diffusion Distance (TSGDD) as a scaled version of the linear distance, with $\alpha$ fixed. Namely, let

$$
\begin{aligned}
\tilde{D}_r^2(G_1, G_2) \quad &= (n_1 n_2)^{-2r} \tilde{D}^2 \left( G_1, G_2 \bigg| \alpha = \left( \frac{n_1}{n_2} \right)^r \right) \\
&= \inf_{P|\mathcal{C}(P)} (n_1 n_2)^{-2r} \left\| \left( \frac{n_1}{n_2} \right)^{-r} P L_1 - \left( \frac{n_1}{n_2} \right)^r L_2 P \right\|_F^2
\end{aligned}
\tag{24}
$$

The intuition for this version of the distance measure is that we are constraining the time dilation, $\alpha$, between $G_1$ and $G_2$ to be a power of the ratio of the two graph sizes. The factor $(n_1 n_2)^{-2r}$ is needed to ensure this version of the distance satisfies the triangle inequality, as seen in Theorem 4.

**Theorem 4**. *The TSGDD, as defined above, satisfies the triangle inequality.*

*Proof.* As above, let $G_1$, $G_2$, $G_3$ be three graphs with $n_i = |G_i|$ and $n_1 \leq n_2 \leq n_3$, and let $L_i$ be the Laplacian of $G_i$. Let $\mathcal{C}(P)$ represent a transitive constraint predicate, also as described previously. Then, for a constant $r \in \mathbb{R}$, we have:

$$\tilde{D}_r(G_1, G_3) =$$
$$\inf_{P|\mathcal{C}(P)} (n_1 n_3)^{-r} \left\| \left(\frac{n_1}{n_3}\right)^{-r} P L_1 - \left(\frac{n_1}{n_3}\right)^r L_3 P \right\|_F$$
$$\leq \inf_{P_{32}, P_{21}|\mathcal{C}(P_{32}P_{21})} (n_1 n_3)^{-r} \left\| \left(\frac{n_1}{n_3}\right)^{-r} P_{32} P_{21} L_1 - \left(\frac{n_1}{n_3}\right)^r L_3 P_{32} P_{21} \right\|_F$$

under the assumption, as in Eq (25), that $\mathcal{C}(P_{32}) \wedge \mathcal{C}(P_{21}) \Rightarrow \mathcal{C}(P_{32}P_{21})$,

$$\tilde{D}_r(G_1, G_3) \leq$$
$$\inf_{P_{32}, P_{21}|\mathcal{C}(P_{32})\wedge\mathcal{C}(P_{21})} (n_1 n_3)^{-r} \left\| \left(\frac{n_1}{n_3}\right)^{-r} P_{32} P_{21} L_1 - \left(\frac{n_1}{n_3}\right)^r L_3 P_{32} P_{21} \right\|_F$$
$$= \inf_{P_{32}, P_{21}|\mathcal{C}(P_{32})\wedge\mathcal{C}(P_{21})} (n_1 n_3)^{-r} \left\| \left(\frac{n_1}{n_3}\right)^{-r} P_{32} P_{21} L_1 - \left(\frac{n_1 n_3}{n_2^2}\right)^r P_{32} L_2 P_{21} \right.$$
$$\left. + \left(\frac{n_1 n_3}{n_2^2}\right)^r P_{32} L_2 P_{21} - \left(\frac{n_1}{n_3}\right)^r L_3 P_{32} P_{21} \right\|_F$$
$$\leq \inf_{P_{32}, P_{21}|\mathcal{C}(P_{32})\wedge\mathcal{C}(P_{21})} (n_1 n_3)^{-r} \left\| \left(\frac{n_1}{n_3}\right)^{-r} P_{32} P_{21} L_1 - \left(\frac{n_1 n_3}{n_2^2}\right)^r P_{32} L_2 P_{21} \right\|_F$$
$$+ (n_1 n_3)^{-r} \left\| \left(\frac{n_1 n_3}{n_2^2}\right)^r P_{32} L_2 P_{21} - \left(\frac{n_1}{n_3}\right)^r L_3 P_{32} P_{21} \right\|_F$$
$$= \inf_{P_{32}, P_{21}|\mathcal{C}(P_{32})\wedge\mathcal{C}(P_{21})} (n_1 n_3)^{-r} \left(\frac{n_3}{n_2}\right)^r \left\| \left(\frac{n_1}{n_2}\right)^{-r} P_{32} P_{21} L_1 - \left(\frac{n_1}{n_2}\right)^r P_{32} L_2 P_{21} \right\|_F$$
$$+ (n_1 n_3)^{-r} \left(\frac{n_1}{n_2}\right)^r \left\| \left(\frac{n_2}{n_3}\right)^{-r} P_{32} L_2 P_{21} - \left(\frac{n_2}{n_3}\right)^r L_3 P_{32} P_{21} \right\|_F$$
$$= \inf_{P_{32}, P_{21}|\mathcal{C}(P_{32})\wedge\mathcal{C}(P_{21})} (n_1 n_2)^{-r} \left\| \left(\frac{n_1}{n_2}\right)^{-r} P_{32} P_{21} L_1 - \left(\frac{n_1}{n_2}\right)^r P_{32} L_2 P_{21} \right\|_F$$
$$+ (n_2 n_3)^{-r} \left\| \left(\frac{n_2}{n_3}\right)^{-r} P_{32} L_2 P_{21} - \left(\frac{n_2}{n_3}\right)^r L_3 P_{32} P_{21} \right\|_F$$

By Lemma 1,

$$
\begin{aligned}
\tilde{D}_r(G_1, G_3) &\leq \inf_{P_{32}, P_{21}|\mathcal{C}(P_{32}) \wedge \mathcal{C}(P_{21})} (n_1 n_2)^{-r} \left\| \left(\frac{n_1}{n_2}\right)^{-r} P_{21} L_1 - \left(\frac{n_1}{n_2}\right)^r L_2 P_{21} \right\|_F \\
&\quad + (n_2 n_3)^{-r} \left\| \left(\frac{n_2}{n_3}\right)^{-r} P_{32} L_2 - \left(\frac{n_2}{n_3}\right)^r L_3 P_{32} \right\|_F \\
&= \inf_{P_{21}|\mathcal{C}(P_{21})} (n_1 n_2)^{-r} \left\| \left(\frac{n_1}{n_2}\right)^{-r} P_{21} L_1 - \left(\frac{n_1}{n_2}\right)^r L_2 P_{21} \right\|_F \\
&\quad + \inf_{P_{32}|\mathcal{C}(P_{32})} (n_2 n_3)^{-r} \left\| \left(\frac{n_2}{n_3}\right)^{-r} P_{32} L_2 - \left(\frac{n_2}{n_3}\right)^r L_3 P_{32} \right\|_F \\
&= \tilde{D}_r(G_1, G_2) + \tilde{D}_r(G_2, G_3)
\end{aligned}
$$

and so

$$
\tilde{D}_r(G_1, G_3) \leq \tilde{D}_r(G_1, G_2) + \tilde{D}_r(G_2, G_3)
$$

for any fixed $r \in \mathbb{R}$.

## 3.3 Sparse-diffusion distance

We introduce the notation $\mathcal{C}(P)$ for a constraint predicate that must be satisfied by prolongation matrix $P$, which is transitive in the sense that:

$$
\mathcal{C}(P_{32}) \wedge \mathcal{C}(P_{21}) \Rightarrow \mathcal{C}(P_{32} P_{21}). \tag{25}
$$

The simplest example is $\mathcal{C}(P) = \mathcal{C}_{\text{orthog}}(P) \equiv (P^T P = I)$. However, sparsity can be introduced in transitive form by $\mathcal{C}(P) = \mathcal{C}_{\text{orthog}}(P) \wedge C_{\text{sparsity}}(P)$ where

$$
\mathcal{C}_{\text{sparsity}}(P) \equiv \left( \max_{i,j} \text{degree}_{i,j}(P) \leq (n_{P\text{coarse}}/n_{P\text{fine}})^s \right)
$$

for some real number $s \geq 0$. This predicate is transitive since

$$
\max_{i,j} \text{degree}_{i,j}(P_{32} P_{21}) \leq \max_{i,j} \text{degree}_{i,j}(P_{32}) \max_{i,j} \text{degree}_{i,j}(P_{21}),
$$

and since $n_2$ cancels out from the numerator and denominator of the product of the fanout bounds. Here, $\text{degree}_{i,j}(M)$ is the total number of nonzero entries in row $i$ or column $j$ of $M$.

This transitive sparsity constraint depends on a power-law parameter $s \geq 0$. When $s = 0$, there is no sparsity constraint.

Another form of sparsity constraints are those which specify a pattern on matrix entries which are allowed to be nonzero. Two simple examples (which are also transitive) are matrices which are constrained to be upper triangular, as well as matrices which are constrained to be of the form $A \otimes B$ where $A$ and $B$ are themselves both constrained to be sparse. More complicated are $n_1 \times n_2$ matrices which are constrained to be banded for some specified pattern of bands: more specifically, that there is a reordering of the rows and columns that the number of diagonal bands (of width 1, slope $\frac{n_1}{n_2}$) with nonzero entries is less than $\left(\frac{n_1}{n_2}\right)^q$ for some $0 \leq q < 1$. For example, linear interpolation matrices between d-dimensional grids, with non-overlapping source regions, follow this constraint.

As a final note on sparsity, we observe that any of the optimizations detailed in this work could also be performed including a sparsity term (for example, the $|\cdot|_1$-norm of the matrix $P$, calculated as $\Sigma_i \Sigma_j |p_{ij}|$ is one possibility, as are terms which penalize $t$ or $\alpha$ far from 1), rather

than explicit sparsity constraints. A potential method of performing this optimization would be to start by optimizing the non-sparse version of the objective function (as detailed in Section 4.1) and then slowly increasing the strength of the regularization term.

## 3.4 Upper bounds for graph products

We now consider the case where we want to compute the distance of two graph box products, i.e. $D(\mathbf{G}_1, \mathbf{G}_2)$ where

$$\mathbf{G}_1 = G_1^{(1)} \square G_1^{(2)} \quad \text{and} \quad \mathbf{G}_2 = G_2^{(1)} \square G_2^{(2)} \tag{26}$$

and

$$P^{(1)} = \arg \inf_{P_c | \mathcal{C}(P_c)} D(G_1^{(1)}, G_2^{(1)} | t_c, \alpha_c, P_c)$$

$$P^{(2)} = \arg \inf_{P_c | \mathcal{C}(P_c)} D(G_1^{(2)}, G_2^{(2)} | t_c, \alpha_c, P_c) \tag{27}$$

are known for some $t_c, \alpha_c$. Previous work [2] included a proof of a similar inequality for the small-$t$ ("linear") case of our objective function.

**Theorem 5**. *Let* $\mathbf{G}_1$ *and* $\mathbf{G}_2$ *be graph box products as described above, and for a graph G let* $L(G)$ *be its Laplacian. For fixed* $t = t_c$, $\alpha = \alpha_c$, $P^{(i)}$ *as given above, for any* $\lambda \in [0, 1]$, *we have*

$$\inf_{P_c | \mathcal{C}(P_c)} D(\mathbf{G}_1, \mathbf{G}_2) \quad \leq$$

$$\lambda \left( \left\| e^{\frac{t_c}{\alpha_c} L(G_1^{(2)})} \right\|_F + \left\| e^{t_c \alpha_c L(G_2^{(2)})} \right\|_F \right) D\left( G_1^{(1)}, G_2^{(1)} | P^{(1)} \right) \tag{28}$$

$$+ (1 - \lambda) \left( \left\| e^{\frac{t_c}{\alpha_c} L(G_1^{(1)})} \right\|_F + \left\| e^{t_c \alpha_c L(G_2^{(1)})} \right\|_F \right) D\left( G_1^{(2)}, G_2^{(2)} | P^{(2)} \right)$$

*where all distances are evaluated at* $t = t_c$, $\alpha = \alpha_c$, *but we have omitted those terms for simplicity of notation.*

*Proof.* For graph products $\mathbf{G}_i$, we have

$$L(\mathbf{G}_i) = L(G_i^{(1)}) \oplus L(G_i^{(2)})$$

$$= (L(G_i^{(1)}) \otimes I_{|L(G_i^{(2)})|}) + (I_{|L(G_i^{(1)})|} \otimes L(G_i^{(2)})) \tag{29}$$

(this fact can be easily verified from the formula for the adjacency matrix of a graph box product, given in the definition in Section 1.3), and so

$$\exp\left[ cL(\mathbf{G}_i) \right] = \exp\left[ c(L(G_i^{(1)}) \otimes I_{|L(G_i^{(2)})|}) + (I_{|L(G_i^{(1)})|} \otimes L(G_i^{(2)})) \right]. \tag{30}$$

Because $A \otimes I_{|B|}$ and $I_{|A|} \otimes B$ commute for any $A$ and $B$,

$$\exp\left[ cL(\mathbf{G}_i) \right] = \exp\left[ c(L(G_i^{(1)}) \otimes I_{|L(G_i^{(2)})|}) \right] \exp\left[ c(I_{|L(G_i^{(1)})|} \otimes L(G_i^{(2)})) \right]$$

$$= \left( \exp\left[ cL(G_i^{(1)}) \right] \otimes I_{|L(G_i^{(2)})|} \right) \left( I_{|L(G_i^{(1)})|} \otimes \exp\left[ cL(G_i^{(2)}) \right] \right) \tag{31}$$

$$= \exp\left[ cL(G_i^{(1)}) \right] \otimes \exp\left[ cL(G_i^{(2)}) \right]$$

We will make the following abbreviations:

$$\mathbf{E}_1 = e^{\frac{t_c}{\alpha_c}L(\mathbf{G}_1)} \qquad E_1^{(1)} = e^{\frac{t_c}{\alpha_c}L(G_1^{(1)})} \qquad E_1^{(2)} = e^{\frac{t_c}{\alpha_c}L(G_1^{(2)})}$$

$$\mathbf{E}_2 = e^{t_c\alpha_c L(\mathbf{G}_2)} \qquad E_2^{(1)} = e^{t_c\alpha_c L(G_2^{(1)})} \qquad E_2^{(2)} = e^{t_c\alpha_c L(G_2^{(2)})}$$

Then,

$$\inf_{P|\mathcal{C}(P)} D(\mathbf{G}_1, \mathbf{G}_2) \quad \leq D(\mathbf{G}_1, \mathbf{G}_2|P^{(1)} \otimes P^{(2)})$$
$$= \left\| (P^{(1)} \otimes P^{(2)})\mathbf{E}_1 - \mathbf{E}_2(P^{(1)} \otimes P^{(2)}) \right\|_F \tag{32}$$

$$= \left\| (P^{(1)} \otimes P^{(2)})(E_1^{(1)} \otimes E_1^{(2)}) - (E_2^{(1)} \otimes E_2^{(2)})(P^{(1)} \otimes P^{(2)}) \right\|_F$$
$$= \left\| (P^{(1)}E_1^{(1)} \otimes P^{(2)}E_1^{(2)}) - (E_2^{(1)}P^{(1)} \otimes E_2^{(2)}P^{(2)}) \right\|_F^2$$
$$= \left\| (P^{(1)}E_1^{(1)} \otimes P^{(2)}E_1^{(2)}) - (P^{(1)}E_1^{(1)} \otimes E_2^{(2)}P^{(2)}) \right.$$
$$\left. + (P^{(1)}E_1^{(1)} \otimes E_2^{(2)}P^{(2)}) - (E_2^{(1)}P^{(1)} \otimes E_2^{(2)}P^{(2)}) \right\|_F \tag{33}$$
$$\leq \left\| (P^{(1)}E_1^{(1)} \otimes P^{(2)}E_1^{(2)}) - (P^{(1)}E_1^{(1)} \otimes E_2^{(2)}P^{(2)}) \right\|_F$$
$$+ \left\| (P^{(1)}E_1^{(1)} \otimes E_2^{(2)}P^{(2)}) - (E_2^{(1)}P^{(1)} \otimes E_2^{(2)}P^{(2)}) \right\|_F$$

$$= \left\| P^{(1)}E_1^{(1)} \otimes (P^{(2)}E_1^{(2)} - E_2^{(2)}P^{(2)}) \right\|_F$$
$$+ \quad \left\| (P^{(1)}E_1^{(1)} - E_2^{(1)}P^{(1)}) \otimes E_2^{(2)}P^{(2)} \right\|_F \tag{34}$$

$$= \left\| P^{(1)}E_1^{(1)} \right\|_F \left\| P^{(2)}E_1^{(2)} - E_2^{(2)}P^{(2)} \right\|_F$$
$$+ \quad \left\| P^{(1)}E_1^{(1)} - E_2^{(1)}P^{(1)} \right\|_F \left\| E_2^{(2)}P^{(2)} \right\|_F. \tag{35}$$

By Lemma 1,

$$\inf_{P|\mathcal{C}(P)} D(\mathbf{G}_1, \mathbf{G}_2) \quad \leq \left\| E_1^{(1)} \right\|_F \left\| P^{(2)}E_1^{(2)} - E_2^{(2)}P^{(2)} \right\|_F$$
$$+ \quad \left\| P^{(1)}E_1^{(1)} - E_2^{(1)}P^{(1)} \right\|_F \left\| E_2^{(2)} \right\|_F. \tag{36}$$

If we instead use $(E_2^{(1)}P^{(1)} \otimes P^{(2)}E_1^{(2)})$ as the cross term in Eq (33), we have

$$\inf_{P} D(\mathbf{G}_1, \mathbf{G}_2) \leq \left\| E_2^{(1)} \right\|_F \left\| P^{(2)}E_1^{(2)} - E_2^{(2)}P^{(2)} \right\|_F$$
$$+ \quad \left\| P^{(1)}E_1^{(1)} - E_2^{(1)}P^{(1)} \right\|_F \left\| E_1^{(2)} \right\|_F \tag{37}$$

A linear combination of these two bounds gives us the desired bound.
This has the additional consequence that

$$\inf_{P_c|\mathcal{C}(P_c)} D(\mathbf{G}_1, \mathbf{G}_2) \quad \leq$$
$$\min\left[ \left( \left\| e^{\frac{t_c}{\alpha_c}L(G_1^{(2)})} \right\|_F + \left\| e^{t_c\alpha_c L(G_2^{(2)})} \right\|_F \right) D\left( G_1^{(1)}, G_2^{(1)}|P^{(1)} \right), \right.$$
$$\left. \left( \left\| e^{\frac{t_c}{\alpha_c}L(G_1^{(1)})} \right\|_F + \left\| e^{t_c\alpha_c L(G_2^{(1)})} \right\|_F \right) D\left( G_1^{(2)}, G_2^{(2)}|P^{(2)} \right) \right] \tag{38}$$

Additionally, if

$$E_i^{(1)} = E_i^{(2)} \text{ for } i \in 1, 2 \quad \text{and} \quad P^{(1)} = P^{(2)}, \tag{39}$$

This reduces further to

$$D(\mathbf{G}_1, \mathbf{G}_2 | P^{(1)} \otimes P^{(1)}) \quad \leq \min(\|E_1^{(1)}\|_F, \|E_2^{(1)}\|_F) \|P^{(1)} E_1^{(1)} - E_2^{(1)} P^{(1)}\|_F \tag{40}$$

and so

$$D(G_1^{(1)} \Box G_1^{(1)}, G_2^{(1)} \Box G_2^{(1)} | t_c, \alpha_c) \\ \leq \min\left(\left\|e^{\frac{t_c}{\alpha_c}L(G_1^{(1)})}\right\|_F, \left\|e^{t_c \alpha_c L(G_2^{(1)})}\right\|_F\right) D\left(G_1^{(1)}, G_2^{(1)} | t_c, \alpha_c\right) \tag{41}$$

An example of such a graph sequence is the sequence of two-dimensional square grids, which are each the box product of two identical one-dimensional grids i.e. path graphs: $Sq_n = Pa_n \Box Pa_n$.

## 3.5 Spectral lower bound

In Theorem 7 we will derive and make use of an upper bound on the graph distance $\tilde{D}(G_1, G_2)$. This upper bound is calculated by constraining the variable $P$ to be not only orthogonal, but also $P = U_2 M U_1^T$ where M is the solution (i.e. "matching", in the terminology of that section) to a Linear Assignment problem with costs given by a function of the eigenvalues of $L(G_1)$ and $L(G_2)$. In this section we derive a similar lower bound on the distance.

Let $G_1$ and $G_2$ be undirected graphs with Laplacians $L_1 = L(G_1)$ and $L_2 = L(G_2)$, and let $\alpha > 0$ be constant. By Eq (52), we have

$$\tilde{D}^2(G_1, G_2) \quad = \inf_{\alpha > 0} \inf_{P^T P = I} \left( \sum_{i=1}^{n_2} \sum_{j=1}^{n_1} p_{ij}^2 \left( \frac{1}{\alpha} \lambda_j^{(1)} - \alpha \lambda_i^{(2)} \right)^2 \right). \tag{42}$$

The following upper bound on $\tilde{D}$ is achieved by constraining $P$ to be not only orthogonal, but related to a constrained matching problem between the two lists of eigenvalues:

$$\tilde{D}^2(G_1, G_2) \leq \inf_{\alpha > 0} \inf_M \quad \left\| \frac{1}{\alpha} M \Lambda_1 - \alpha \Lambda_2 M \right\|_F^2$$

$$\text{subject to} \qquad \sum_{i=1}^{n_2} m_{ij} \leq 1, \qquad j = 1 \dots n_1$$

$$\sum_{j=1}^{n_1} m_{ij} \leq 1, \qquad i = 1 \dots n_2 \tag{43}$$

$$m_{ij} \geq 0 \qquad i = 1 \dots n_2, j = 1 \dots n_1,$$

where $\Lambda_1$ and $\Lambda_2$ are diagonal matrices of the eigenvalues of $L_1$ and $L_2$ respectively. Here we used the explicit map $\tilde{P} = U_2^T P U_1$ as a change of basis; we then converted the constraints on $P$ into equivalent constraints on $\tilde{P}$, and imposed additional constraints so that the resulting optimization (a linear assignment problem) is an upper bound. See the proof of Theorem 7 for the details of this derivation. We show in this section that a less constrained assignment problem is a lower bound on $\tilde{D}^2$. We do this by computing the same mapping $\tilde{P} = U_2^T P U_1$ and then dropping some of the constraints on $\tilde{P}$ (which is equivalent to dropping constraints on $P$,

yielding a lower bound). The constraint $P^T P = I$ is the conjunction of $n_1^2$ constraints on the column vectors of $P$: if $\mathbf{p}_i$ is the $i$th column of $P$, then $P^T P = I$ is equivalent to:

$$\mathbf{p}_i \cdot \mathbf{p}_i = 1 \qquad \forall i = 1 \ldots n_1 \qquad\qquad (44)$$

$$\mathbf{p}_i \cdot \mathbf{p}_i = 0 \qquad \forall i = 1 \ldots n_1, j = 1 \ldots i-1, i+1 \ldots n_1, \qquad (45)$$

If we discard the constraints in Eq (45), we are left with the constraint that every column of $p$ must have unit norm.

Construct the "spectral lower bound matching" matrix $P^{(\mathrm{SLB})}$ as follows:

$$P_{i,j}^{(\mathrm{SLB})} = \begin{cases} 1 & \text{if } i = \arg\min_k \left( \frac{1}{\alpha}\lambda_j^{(1)} - \alpha\lambda_k^{(k)} \right)^2 \\ 0 & \text{otherwise.} \end{cases} \qquad (46)$$

For any $\alpha$, this matrix is the solution to a matching problem (less constrained than the original optimization over all $P$) where each $\lambda_j^{(1)}$ is assigned to the closest $\lambda_i^{(2)}$, allowing collisions. It clearly satisfies the constraints in Eq (44), but may violate those in Eq (45). Thus, we have

$$
\begin{aligned}
\tilde{D}^2(G_1, G_2) &= \inf_{\alpha>0} \inf_{P^T P=I} \left( \sum_{i=1}^{n_2} \sum_{j=1}^{n_1} p_{ij}^2 \left( \frac{1}{\alpha}\lambda_j^{(1)} - \alpha\lambda_i^{(2)} \right)^2 \right). \\
&\geq \tilde{D}^2\left(G_1, G_2 \middle| P^{(\mathrm{SLB})}\right)
\end{aligned}
\qquad (47)
$$

Various algorithms exist to rapidly find the member of a set of points which is closest to some reference point (for example, KD-Trees [29]). For any $\alpha$, the spectral lower bound can be calculated by an outer loop over alpha and an inner loop which applies one of these methods. We do not consider joint optimization of the lower bound over $P$ and $\alpha$ in this work.

### 3.6 Regularized distance

We can add a regularization term to the graph diffusion distance, as follows: define

$$
\begin{aligned}
D_{\mathrm{reg}}(G_1, G_2) &= \sup_t \inf_{P|\mathcal{C}(P)} \inf_{\alpha>0} \{ \left\| Pe^{\frac{t}{\alpha}L_1} - e^{t\alpha L_2}P \right\|_F + \left\| e^{\frac{t}{\alpha}L_1} - e^{tL_1} \right\|_F \\
&\qquad\qquad + \left\| e^{tL_2}P - e^{t\alpha L_2}P \right\|_F \}
\end{aligned}
$$

We can show analytically that this distance satisfies the triangle inequality:

**Theorem 6**. $D_{reg}$ *satisfies the triangle inequality.*

*Proof.* For graphs $G_1$, $G_2$, $G_3$ and Laplacians $L_1$, $L_2$, $L_3$, for any fixed $t \geq 0$, we have:

$$
\begin{aligned}
D_{\mathrm{reg}}(G_1, G_3|t) &= \inf_{P|\mathcal{C}(P)} \inf_{\alpha>0} \{ \left\| Pe^{\frac{t}{\alpha}L_1} - e^{t\alpha L_3}P \right\|_F + \left\| e^{\frac{t}{\alpha}L_1} - e^{tL_1} \right\|_F \\
&\qquad\qquad + \left\| e^{tL_3}P - e^{t\alpha L_3}P \right\|_F \} \\
&\leq D_{\mathrm{reg}}(G_1, G_3|t, \alpha=1) \\
&= \inf_{P|\mathcal{C}(P)} \{ \left\| Pe^{tL_1} - e^{tL_3}P \right\|_F + \left\| e^{tL_1} - e^{tL_1} \right\|_F \\
&\qquad\qquad + \left\| e^{tL_3}P - e^{tL_3}P \right\|_F \} \\
&= \inf_{P|\mathcal{C}(P)} \left\| Pe^{tL_1} - e^{tL_3}P \right\|_F
\end{aligned}
$$

Suppose that

$$\alpha_{32}, P_{32} \quad = \arg\inf_{a>0} \inf_{P|\mathcal{C}(P)} \left\{ \left\| Pe^{\frac{t}{2}L_2} - e^{t\alpha L_3}P \right\|_F + \left\| e^{\frac{t}{2}L_2} - e^{tL_2} \right\|_F \right.$$
$$\left. + \left\| e^{tL_3}P - e^{t\alpha L_3}P \right\|_F \right\}$$

$$\alpha_{21}, P_{21} \quad = \arg\inf_{a>0} \inf_{P|\mathcal{C}(P)} \left\{ \left\| Pe^{\frac{t}{2}L_1} - e^{t\alpha L_2}P \right\|_F + \left\| e^{\frac{t}{2}L_1} - e^{tL_1} \right\|_F \right.$$
$$\left. + \left\| e^{tL_2}P - e^{t\alpha L_2}P \right\|_F \right\}$$

Then,

$$\inf_{P|\mathcal{C}(P)} \left\| Pe^{tL_1} - e^{tL_3}P \right\|_F \quad \leq \left\| P_{32}P_{21}e^{tL_1} - e^{tL_3}P_{32}P_{21} \right\|_F$$

$$\inf_{P|\mathcal{C}(P)} \left\| Pe^{tL_1} - e^{tL_3}P \right\|_F \quad \leq \left\| P_{32}P_{21}e^{tL_1} - P_{32}P_{21}e^{\frac{t}{\alpha_{21}}L_1} + P_{32}P_{21}e^{\frac{t}{\alpha_{21}}L_1} \right.$$

$$- P_{32}e^{t\alpha_{21}L_2}P_{21} + P_{32}e^{t\alpha_{21}L_2}P_{21} - P_{32}e^{tL_2}P_{21}$$

$$+ P_{32}e^{tL_2}P_{21} - P_{32}e^{\frac{t}{\alpha_{32}}L_2}P_{21} + P_{32}e^{\frac{t}{\alpha_{32}}L_2}P_{21}$$

$$- e^{t\alpha_{32}L_3}P_{32}P_{21} + e^{t\alpha_{32}L_3}P_{32}P_{21} - e^{tL_3}P_{32}P_{21} \Big\|_F$$

$$\leq \left\| P_{32}P_{21}e^{tL_1} - P_{32}P_{21}e^{\frac{t}{\alpha_{21}}L_1} \right\|_F$$

$$+ \left\| P_{32}P_{21}e^{\frac{t}{\alpha_{21}}L_1} - P_{32}e^{t\alpha_{21}L_2}P_{21} \right\|_F$$

$$+ \left\| P_{32}e^{t\alpha_{21}L_2}P_{21} - P_{32}e^{tL_2}P_{21} \right\|_F$$

$$+ \left\| P_{32}e^{tL_2}P_{21} - P_{32}e^{\frac{t}{\alpha_{32}}L_2}P_{21} \right\|_F$$

$$+ \left\| P_{32}e^{\frac{t}{\alpha_{32}}L_2}P_{21} - e^{t\alpha_{32}L_3}P_{32}P_{21} \right\|_F$$

$$+ \left\| e^{t\alpha_{32}L_3}P_{32}P_{21} - e^{tL_3}P_{32}P_{21} \right\|_F$$

by Lemma 1,

$$\inf_{P|\mathcal{C}(P)} \left\| Pe^{tL_1} - e^{tL_3}P \right\|_F \quad \leq \left\| e^{tL_1} - e^{\frac{t}{\alpha_{21}}L_1} \right\|_F + \left\| P_{21}e^{\frac{t}{\alpha_{21}}L_1} - e^{t\alpha_{21}L_2}P_{21} \right\|_F$$

$$+ \left\| e^{t\alpha_{21}L_2}P_{21} - e^{tL_2}P_{21} \right\|_F$$

$$+ \left\| e^{tL_2} - e^{\frac{t}{\alpha_{32}}L_2} \right\|_F + \left\| P_{32}e^{\frac{t}{\alpha_{32}}L_2} - e^{t\alpha_{32}L_3}P_{32} \right\|_F$$

$$+ \left\| e^{t\alpha_{32}L_3}P_{32} - e^{tL_3}P_{32} \right\|_F$$

$$= D_{\text{reg}}(G_1, G_2|t = c) + D_{\text{reg}}(G_2, G_3|t = c)$$

Since this is true for any fixed $t$, let

$$t^* = \arg\sup_t D_{\text{reg}}(G_1, G_3|t).$$

Then

$$
\begin{aligned}
D_{\mathrm{reg}}(G_1, G_3) \quad &= \sup_{c} D_{\mathrm{reg}}(G_1, G_3 | t) \\
&= D_{\mathrm{reg}}(G_1, G_3 | t^*) \\
&\leq D_{\mathrm{reg}}(G_1, G_2 | t^*) + D_{\mathrm{reg}}(G_2, G_3 | t = t^*) \\
&\leq \sup_{t_{21}} D_{\mathrm{reg}}(G_1, G_2 | t_{21}) + \sup_{t_{32}} D_{\mathrm{reg}}(G_2, G_3 | t_{32}) \\
&= D_{\mathrm{reg}}(G_1, G_2) + D_{\mathrm{reg}}(G_2, G_3)
\end{aligned}
$$

We can construct a similar regularized version of the linear objective function:

$$
\tilde{D}_{\mathrm{reg}}(G_1, G_2) \quad = \left\| \frac{1}{\alpha} P L_1 - \alpha L_2 P \right\| + \left\| \frac{1}{\alpha} L_1 - L_1 \right\| + \left\| P L_2 - \alpha L_2 P \right\|
$$

The term "regularized" here refers to the fact that the additional terms included in $D_{\mathrm{reg}}$ and $\tilde{D}_{\mathrm{reg}}$ penalize $\alpha$ distorting the respective Laplacians far from their original values. In practice, many of the theoretical guarantees provided earlier in this manuscript may not apply to optimization of the augmented objective function. Hence, a major area of future work will be modification of our optimization procedure to compute this form of distance.

### 3.7 Theory summary

Triangle inequalities are proven for some members of the proposed family of graph distortion or "distance" measures, including infinitesimal and finite diffusion time, a power law for sparsity, and/or a power law for the time scaling factor between coarse and fine scales. However, the case of an optimal (not power law) time conversion factor $\alpha$ needs to be investigated by numerical experiment, and that requires new algorithms, introduced in Section 4. We also show that in the case of distances between graph box products, optimization over $P$ for the product graphs is bounded above by a monotonic function of the optimum over the component graphs.

### 3.8 Summary of distance metric versions

Table 1 summarizes the variants of our distance metric.

## 4 Numerical methods for optimal time conversion, *α*

Optimizing the $\alpha$ parameter for conversion between coarse and fine time scales in the proposed family of graph distance measures, in addition to optimizing the prolongation matrix $P$ under transitive constraints $\mathcal{C}(P)$, is a nontrivial numerical problem that in our experience seems to require new methods. We develop such methods here and apply them to investigate the resulting graph "distance" measure in the next section.

### 4.1 Algorithm development

In this section, we describe the algorithm used to calculate upper bounds on graph distances as the joint optima (over $P$, $t$, and $\alpha$) of the distance Eqs 8 and 9, under orthogonality constraints only, i.e. the case $\mathcal{C}(P) = \{P | P^T P = I\}$. At the core of both algorithms is a subroutine to solve the Linear Assignment Problem (LAP—see Eq (5)) repeatedly, in order to find the subpermutation matrix which is optimal at a particular value of $\alpha$. Namely, we are interested in

calculating $\tilde{D}$ as

$$\tilde{D}(G_1, G_2) = \min_\alpha f(\alpha) \quad \text{where} \quad f(\alpha) = \inf_{P|P^T P = I} \left\| \frac{1}{\alpha} PL(G_1) - \alpha L(G_2) P \right\| \tag{48}$$

which, for orthogonality or any other compact constraint

$$= \min_{P|P^T P = I} \left\| \frac{1}{\alpha} PL(G_1) - \alpha L(G_2) P \right\|.$$

However, we have found that the unique structure of this optimization problem admits a specialized procedure which is faster and more accurate than nested univariate optimization of $\alpha$ and $t$ (where each innermost function evaluation consists of a full optimization over $P$ at some $t, \alpha$). We first briefly describe the algorithm used to find the optimal $P$ and $\alpha$ for $\tilde{D}^2$. The formal description of the algorithm is given by Algorithm 1. In both cases, we reduce the computational complexity of the optimization over $P$ by imposing the additional constraint that $P$ must be a subpermutation matrix when rotated into the spectral basis (we define subpermutations in the proof of Theorem 7). This constraint is compatible with the orthogonality constraint (all subpermutation matrices are orthogonal, but not vice versa). The tradeoff of this reduction of computational complexity is that we can only guarantee that our optima are upper bounds of the optima over all orthogonal $P$. However, in practice, this bound seems to be tight: we have yet to find an example where orthogonally-constrained optimization was able to improve in objective function value over optimization constrained to subpermutation matrices. Therefore, we shall for the remainder of this paper refer to the optima calculated as distance values, when strictly they are distance upper bounds. We also note here that a distance lower bound is also possible to calculate by *relaxing* the constraints in $\mathcal{C}(P)$ (for instance, by replacing the optimization over all $P$ with a less constrained matching problem—see Section 3.5).

### 4.1.1 Optimization of $\tilde{D}^2$.

**Algorithm 1** Abbreviated pseudocode for the algorithm described in Section 4.1.1, for computing $\inf_{P,\alpha} \tilde{D}^2$.

```
1: procedure D-TILDE(L₁, L₂, α_low, α_high.)
2:    Compute λ⁽¹⁾, λ⁽²⁾ as the eigenvalues of L₁ and L₂.
3:    Compute, by optimizing a linear assignment, M_low and M_high as the
      optimal matchings at α_low, α_high respectively. Initialize the list of
      optimal matchings as {M_low, M_high}.
4:    Until the current list of matchings is not expanded in the follow-
      ing step, or the entire interval [α_low, α_high] is marked as explored:
5:       Attempt to expand the list of optimal matchings by solving a
      linear assignment problem at the α where the cost curves of two match-
      ings (currently in the list) intersect. If no better assignment
      exists, then mark the interval covered by those matchings as explored,
      as guaranteed by Theorem 9.
6:    Return the lowest-cost M and its optimal α.
7: end procedure
```

Joint optimization of $\tilde{D}^2$ over $\alpha$ and $P$ is a nested optimization problem (see [30] and [31] for a description of nested optimization), with potential combinatorial optimization over $P$ dependent on each choice of $\alpha$. Furthermore, the function $f(\alpha) = \inf_{P|\mathcal{C}(P)} \tilde{D}^2(G_1, G_2|\alpha)$ is both multimodal and continuous but with in general discontinuous derivative (See Fig 4). Univariate optimization procedures such as Golden Section Search result in many loops of some procedure to optimize over $P$, which in our restricted case must each time compute a full solution

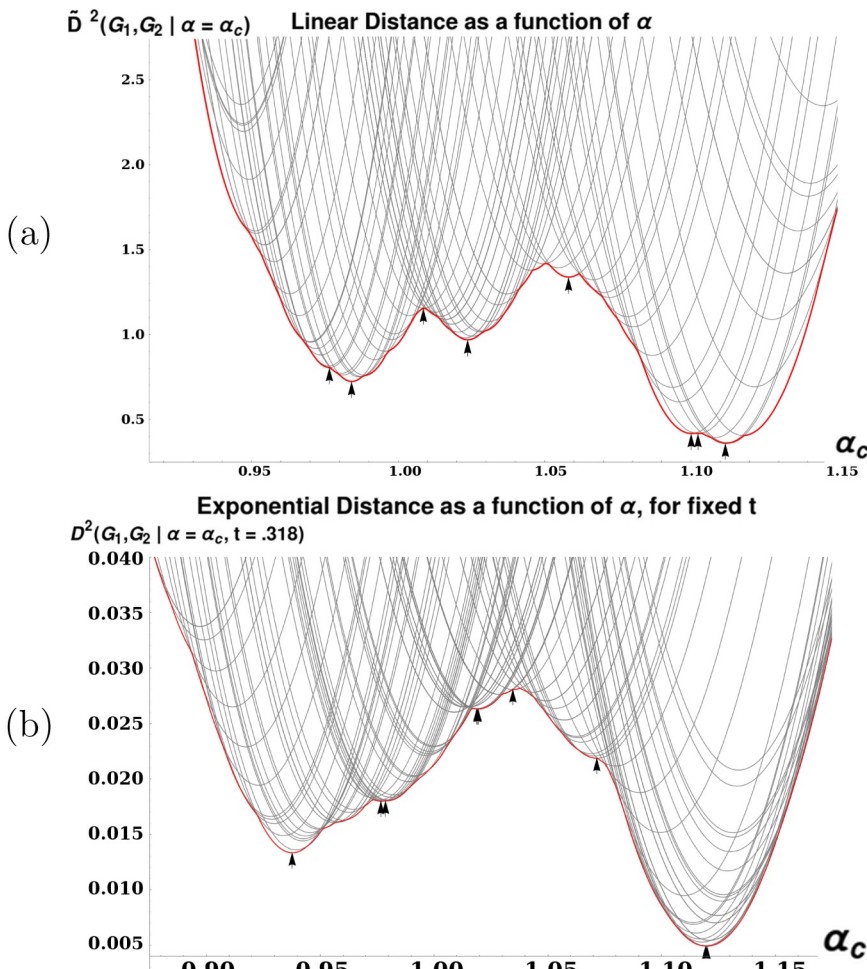

**Fig 4. Two plots demonstrating characteristics of distance calculation between a $(7 \times 7)$ grid and an $(8 \times 8)$ grid.**
(a): Plot illustrating the discontinuity and multimodality of the linear version of distance. Each gray curve represents a function $f_{P_c}(\alpha_c) = \tilde{D}^2(\mathrm{Sq}_7, \mathrm{Sq}_8 | \alpha_c, P_c)$. The thicker curve is the lower convex hull of the thinner curves as a function of $\alpha$, that is: $f(\alpha_c) = \inf_{P|\mathcal{C}(P)} \tilde{D}^2(\mathrm{Sq}_7, \mathrm{Sq}_8 | \alpha_c)$. We see that $f(\alpha)$ is continuous, but has discontinuous slope, as well as several local optima (marked by arrowheads). These properties make $\tilde{D}$ difficult to optimize, necessitating the development of Algorithm 1. (b): As in (a), but with $D^2(\mathrm{Sq}_7, \mathrm{Sq}_8 | t = .318)$ plotted instead of $\tilde{D}^2$. This $t$ value is the location of the maximum in Fig 3.

to a LAP with $n_2 \times n_1$ weights. In our experience, this means that these univariate methods have a tendency to get stuck in local optima. We reduce the total number of calls to the LAP solver, as well as the size of the LAPs solved, by taking advantage of several unique properties of the optimization as a function of $\alpha$. When the optimal $P^{(1)}$ and $P^{(2)}$ are known for $\alpha_1$ and $\alpha_2$, then for any $\alpha_c$ such that $\min(\alpha_1, \alpha_2) \leq \alpha_c \leq \max(\alpha_1, \alpha_2)$, the optimal $P^{(c)}$ at $\alpha_c$ must satisfy: $P_{ij}^{(1)} = 1 \wedge P_{ij}^{(2)} = 1 \Rightarrow P_{ij}^{(c)} = 1$ (see Theorem 9). Thus, the optimization over $P$ at $\alpha_c$ is already partially solved given the solutions at $\alpha_1$ and $\alpha_2$, and so we need only re-compute the remaining (smaller) subproblem on the set of assignments where $P^{(1)}$ and $P^{(2)}$ disagree. This has two consequences for our search over $\alpha$: First, the size of LAP problems which must be solved at each step decreases over time (as we find $P$-optima for a denser and denser set of $\alpha$). Secondly, these theoretical guarantees mean that we can mark intervals of $\alpha$-values as being explored

(meaning we have provably found the *P* which are optimal over the interval) and thus do not have to perform the relatively expensive optimization over *P* for any $\alpha$ in that interval.

### 4.1.2 Optimization of $D^2$.

**Algorithm 2** Abbreviated pseudocode for the algorithm described in Section 4.1.2, for computing $\sup_t \inf_{P,\alpha} D^2$.

```
1: procedure D(L₁, L₂, α_low, α_high, step size ε)
2:    Compute λ⁽¹⁾, λ⁽²⁾ as the eigenvalues of L₁ and L₂.
3:    Solve the Linear Version of the problem using Algorithm 1, obtain-
ing α*, M*. According to the argument presented in the definition of
linear distance (Eq 9) this solution holds for very small t. Keep the
entire frontier of matchings found during the execution of Algorithm
1. Set t = 0, d(0) = D(G₁, G₂|α*, M*, t)
4:    Until d(t + ε)<d(t):
5:    t = t + ε
6:      Use the linear algorithm with e^{tL₁} and e^{tL₂} as the input matrices,
initializing the list of matchings with those found at the previous t.
7:      Set d(t) = D(G₁, G₂|α*, M*, t) where α*, M* are the optima from
the previous step.
8:    Return the max_t d(t).
9: end procedure
```

Many of the theoretical guarantees underlying our algorithm for computing $\tilde{D}^2$ no longer hold for the exponential version of the distance. We adapt our linear-version procedure into an algorithm for computing this version, with the caveat that the lack of these guarantees means that our upper bound on the exponential version may be looser than that on the linear version. It is still clearly an upper bound, since the $\alpha$ and *P* found by this procedure satisfy the given constraints $\alpha > 0$ and $P^T P = I$. In particular, we have observed cases where the exponential-distance analog of Theorem 9 would not hold, meaning we cannot rule out $\alpha$-intervals as we can in the linear version. Thus, this upper bound may be looser than the that computed for the linear objective function.

For the exponential version of the algorithm, we first compute the list of optimal *P* for the linear version, assuming (since $e^{tL} \approx I + L$ for very small *t*) that this is also the list of optimal *P* for the exponential version of the objective function at some low *t*. We proceed to increment *t* with some step size $\Delta t$, in the manner of a continuation method [32]. At each new *t* value, we search for new optimal *P* along the currently known frontier of optima as a function of $\alpha$. When a new *P* is found as the intersection of two known $P_i, P_{i+1}$, it is inserted into the list, which is kept in order of increasing $\alpha$. For each *P* in this frontier, we find the optimal $\alpha$, keeping *P* and *t* constant. Assuming $\inf_P \inf_\alpha D^2(G_1, G_2|t_c)$ is unimodal as a function of $t_c$, we increase $t_c$ until $\inf_P \inf_\alpha D^2(G_1, G_2|t_c) \geq \inf_P \inf_\alpha D^2(G_1, G_2|t_c + \Delta t)$, storing all *P* matrices found as optima at each $t_c$ value. *P* which were on the lower convex hull at some prior value of *t* but not the current value are retained, as they may regain optimality for some $\alpha$-range at a future value of *t* (we have observed this, in practice). For this list $P_1, P_2 \ldots P_m$, we then compute $\sup_t \inf_\alpha \inf_i D^2(G_1, G_2|P_i)$. Since the exponential map is continuous, and we are incrementing *t* by very small steps, we also propose the further computational shortcut of storing the list of optimal $\alpha$ at time *t* to use as starting points for the optimization at $t + \Delta t$. In practice, this made little difference in the runtime of our optimization procedure.

## 4.2 Algorithm correctness proof

**Theorem 7**. *For any two graphs $G_1$ and $G_2$ with Laplacians $L(G_1)$ and $L(G_2)$, for fixed $\alpha$, the optimization over P given in the innermost loop of* Eq 9 *is upper bounded by a Linear Assignment Problem as defined in* Eq (5). *This LAP is given by taking R to be the eigenvalues $\lambda_j^{(1)}$ of $L(G_1)$*

and $S$ to be the eigenvalues $\lambda_i^{(2)}$ of $L(G_2)$, *with the cost of a pair (equivalently, one entry of the cost matrix C) given by*

$$C_{ij} = c(s_i, r_j) = c\left(\lambda_i^{(2)}, \lambda_j^{(1)}\right) = \left(\frac{1}{\alpha}\lambda_j^{(1)} - \alpha\lambda_i^{(2)}\right)^2 \tag{49}$$

*Proof.* $L(G_1)$ and $L(G_2)$ are both real symmetric matrices, so they may be diagonalized as $L(G_i) = U_i\Lambda_i U_i^T$, where the $U_i$ are rotation matrices, and the $\Lambda_i$ are diagonal matrices with the eigenvalues $\lambda_1^{(i)}, \lambda_2^{(i)} \ldots \lambda_{n_i}^{(i)}$ along the diagonal. Because the Frobenius norm is invariant under rotation, we have:

$$
\begin{aligned}
\tilde{D}^2(G_1, G_2) \quad &= \inf_{\alpha > 0} \inf_{P^T P = I} \left\| \frac{1}{\alpha} PL(G_1) - \alpha L(G_2)P \right\|_F^2 \\
&= \inf_{\alpha > 0} \inf_{P^T P = I} \left\| \frac{1}{\alpha} U_2^T PL(G_1)U_1 - \alpha U_2^T L(G_2)PU_1 \right\|_F^2 \\
&= \inf_{\alpha > 0} \inf_{P^T P = I} \left\| \frac{1}{\alpha} U_2^T PU_1\Lambda_1 U_1^T U_1 - \alpha U_2^T U_2\Lambda_2 U_2^T PU_1 \right\|_F^2 \\
&= \inf_{\alpha > 0} \inf_{P^T P = I} \left\| \frac{1}{\alpha} U_2^T PU_1\Lambda_1 - \alpha\Lambda_2 U_2^T PU_1 \right\|_F^2.
\end{aligned}
\tag{50}
$$

Because the $U_i$ are orthogonal, the transformation $\tilde{P} = U_2^T PU_1$ preserves orthogonality, so

$$
\begin{aligned}
\tilde{D}^2(G_1, G_2) \quad &= \inf_{\alpha > 0} \inf_{P^T P = I} \left\| \frac{1}{\alpha} P\Lambda_1 - \alpha\Lambda_2 P \right\|_F^2 \\
&= \inf_{\alpha > 0} \inf_{P^T P = I} \left\| \frac{1}{\alpha}\Lambda_1 \right\|_F^2 + \left\| \alpha\Lambda_2 P \right\|_F^2 - 2\mathrm{Tr}[P^T\Lambda_2 P\Lambda_1] \\
&= \inf_{\alpha > 0} \inf_{P^T P = I} \left( \mathrm{Tr}\left[\frac{1}{\alpha^2}\Lambda_1^2\right] + \mathrm{Tr}[\alpha^2 P^T\Lambda_2^2 P] - 2\mathrm{Tr}[P^T\Lambda_2 P\Lambda_1] \right)
\end{aligned}
$$

writing $P = [p_{ij}]$,

$$
\begin{aligned}
\tilde{D}^2(G_1, G_2) \quad &= \inf_{\alpha > 0} \inf_{P^T P = I} \left( \frac{1}{\alpha^2}\sum_{j=1}^{n_1}\lambda_j^{(1)^2} + \alpha^2\sum_{i=1}^{n_2}\sum_{j=1}^{n_1}p_{ij}^2\lambda_i^{(2)^2} \right. \\
&\qquad\qquad\qquad\left. - 2\sum_{i=1}^{n_2}\sum_{j=1}^{n_1}p_{ij}^2\lambda_i^{(2)}\lambda_j^{(1)} \right)
\end{aligned}
\tag{51}
$$

$$
\begin{aligned}
&= \inf_{\alpha > 0} \inf_{P^T P = I} \left( \sum_{i=1}^{n_2}\sum_{j=1}^{n_1}p_{ij}^2\left(\frac{1}{\alpha^2}\lambda_j^{(1)^2} - 2\lambda_i^{(2)}\lambda_j^{(1)} + \alpha^2\lambda_i^{(2)^2}\right) \right) \\
&= \inf_{\alpha > 0} \inf_{P^T P = I} \left( \sum_{i=1}^{n_2}\sum_{j=1}^{n_1}p_{ij}^2\left(\frac{1}{\alpha}\lambda_j^{(1)} - \alpha\lambda_i^{(2)}\right)^2 \right)
\end{aligned}
\tag{52}
$$

For any given $\alpha$,

$$
\inf_{P^T P = I} \left( \sum_{i=1}^{n_2}\sum_{j=1}^{n_1}p_{ij}^2\left(\frac{\lambda_j^{(1)}}{\alpha} - \alpha\lambda_i^{(2)}\right)^2 \right) \leq \inf_{\tilde{P}|\mathrm{sub}(\tilde{P})} \left( \sum_{i=1}^{n_2}\sum_{j=1}^{n_1}\tilde{p}_{ij}^2\left(\frac{\lambda_j^{(1)}}{\alpha} - \alpha\lambda_i^{(2)}\right)^2 \right) \quad,
$$

where subperm($\tilde{P}$) could be any other condition more strict than the constraint $P^T P = I$. Here we take this stricter constraint to be the condition that $\tilde{P}$ is a *subpermutation matrix*: an orthogonal matrix (i.e. $\tilde{P}^T \tilde{P} = I$) for which $\tilde{P} \in \{0, 1\}^{n_2 \times n_1}$. Equivalently, a subpermutation matrix is a $\{0, 1\}$-valued matrix $[\tilde{p}_{ij}]$ such that for each $i \in \{1, \ldots n_1 \leq n_2\}$, exactly one $j \in \{1, \ldots n_2 \geq n_1\}$ takes the value 1 rather than 0 (so $\sum_{j=1}^{n_2} \tilde{P}_{ji} = 1$), and for each $j \in \{1, \ldots n_2 \geq n_1\}$, either zero or one $i \in \{1, \ldots n_1 \leq n_2\}$ takes the value 1 rather than 0 (so $\sum_{i=1}^{n_1} \tilde{P}_{ji} \leq 1$).

Furthermore, this optimization is exactly a linear assignment problem of eigenvalues of $L(G_1)$ to $L(G_2)$, with the cost of a pair $(\lambda_j^{(1)}, \lambda_i^{(2)})$ given by

$$c\left(\lambda_j^{(1)}, \lambda_i^{(2)}\right) = \left(\frac{1}{\alpha}\lambda_j^{(1)} - \alpha\lambda_i^{(2)}\right)^2$$

Note also that the same argument applies to the innermost two optimizations of the calculation of $D^2$ (the exponential version of the diffusion distance) as well as $D_r^2$. In the $D^2$ case the entries of the cost matrix are instead given by

$$c\left(\lambda_j^{(1)}, \lambda_i^{(2)}\right) = \left(e^{\frac{1}{\alpha}\lambda_j^{(1)}} - e^{\alpha\lambda_i^{(2)}}\right)^2$$

If we instead loosen the constraints on $P$, we can calculate a lower bound on the distance. See Appendix 3.5 for lower bound details.

Recall that our definition of a 'matching' in Section 1.3 was a $P$ matrix representing a particular solution to the linear assignment problem with costs given as in Eq (49). For given $G_1$, $G_2$, and some matching $M$, let

$$f_M(\alpha) = \tilde{D}^2(G_1, G_2 | \alpha, U_2^T M U_1) \tag{53}$$

where $U_1$, $U_2$ diagonalize $L_1$ and $L_2$ as in Eq (50).

**Lemma 8.** *For two unique matchings $M_1$ and $M_2$ (for the same $G_1$, $G_2$) the equation $f_{M_1}(\alpha) - f_{M_2}(\alpha) = 0$ has at most one real positive solution in $\alpha$. This follows from the fact that when $P$ and $t$ are fixed, the objective function is a rational function in $\alpha$ (see Eq (51)), with a quadratic numerator and an asymptote at $\alpha = 0$.*

*Proof.* By Eq (51), we have

$$f_{M_1}(\alpha) - f_{M_2}(\alpha) =$$
$$\left(\frac{1}{\alpha^2}\sum_{j=1}^{n_1}\lambda_j^{(1)^2} + \alpha^2\sum_{i=1}^{n_2}\sum_{j=1}^{n_1}[M_1]_{ij}^2\lambda_i^{(2)^2} - 2\sum_{i=1}^{n_2}\sum_{j=1}^{n_1}[M_1]_{ij}^2\lambda_i^{(2)}\lambda_j^{(1)}\right) \tag{54}$$

$$-\left(\frac{1}{\alpha^2}\sum_{j=1}^{n_1}\lambda_j^{(1)^2} + \alpha^2\sum_{i=1}^{n_2}\sum_{j=1}^{n_1}[M_2]_{ij}^2\lambda_i^{(2)^2} - 2\sum_{i=1}^{n_2}\sum_{j=1}^{n_1}[M_2]_{ij}^2\lambda_i^{(2)}\lambda_j^{(1)}\right) \tag{55}$$

$$= \alpha^2\left(\sum_{i=1}^{n_2}\sum_{j=1}^{n_1}[M_1]_{ij}^2\lambda_i^{(2)^2} - \sum_{i=1}^{n_2}\sum_{j=1}^{n_1}[M_2]_{ij}^2\lambda_i^{(2)^2}\right) \tag{56}$$

$$+\left(2\sum_{i=1}^{n_2}\sum_{j=1}^{n_1}[M_2]_{ij}^2\lambda_i^{(2)}\lambda_j^{(1)} - 2\sum_{i=1}^{n_2}\sum_{j=1}^{n_1}[M_1]_{ij}^2\lambda_i^{(2)}\lambda_j^{(1)}\right) \tag{57}$$

Abbreviating the sums, we have

$$\alpha^2(A_1 - A_2) + (C_2 - C_1) = 0 \tag{58}$$

and so

$$\alpha = \pm\sqrt{\frac{C_2 - C_1}{A_1 - A_2}} \tag{59}$$

Since $A_1$, $A_2$, $C_1$, $C_2$ are all nonnegative reals, at most one of these roots is positive.

We will say that a matching $M$ "assigns" $j$ to $i$ if and only if $M_{ij} = 1$.

**Theorem 9**. *If two matchings $M_1$ and $M_3$ which yield optimal upper bounds for the linear distance $\tilde{D}^2$ (at $\alpha_1 \leq \alpha$ and $\alpha_3 \geq \alpha$ respectively) agree on a set of assignments, then the optimal $M$ at $\alpha$ must also agree with that set of assignments.*

*Proof.* We need the following lemmas:

**Lemma 10**. *If an optimal matching assigns i to m(i) (so that eigenvalue $\lambda_i^{(1)}$ of $G_1$ is paired with $\lambda_{f(i)}^{(2)}$ of $G_2$ in the sum of costs* Eq (49)*), then the sequence $m(1), m(2), \ldots m(n_1)$ is monotonic increasing.*

*Proof.* This follows from the fact that the two sequences of eigenvalues are monotonic non-decreasing, so if there's a 'crossing' ($i_1 < i_2$ but $m(i_2) < m(i_1)$) then the new matching obtained by uncrossing those two pairs (performing a 2-opt step as defined in [33]) has strictly lesser objective function value. Hence an optimal matching can't contain any such crossings.

**Lemma 11**. *For all positive real $\alpha^* \geq \epsilon > 0$, let $M_1$ be an optimal matching at $\alpha^* - \epsilon$ and $M_2$ be optimal at $\alpha^* + \epsilon$. For $1 \leq i \leq n_1$, let $s_1(i)$ and $s_2(i)$ be the indices of $\lambda^{(2)}$ paired with i in $M_1$ and $M_2$, respectively. Then for all i, $s_1(i) \leq s_2(i)$.*

*Proof.* Define a "run" for $s_1, s_2$ as a sequence of consecutive indices $l, l+1, \ldots l+k$ in $[1, n_1]$ such that for any $l, l+1$: $\min(s_1(l+1), s_2(l+1)) < \max(s_1(l), s_2(l))$. The following must be true about a "run":

1. Within a run, either $s_1(l) < s_2(l)$ or $s_1(l) > s_2(l)$ for all $l$. Otherwise, we have one or more crossings (as in Lemma 10): for some $l$ we have $s_1(l) > s_1(l+1)$ or $s_2(l) > s_2(l+1)$. Any crossing may be uncrossed for a strictly lower objective function value—violating optimality of $M_1$ or $M_2$.

2. Any pair of matchings as defined above consists of a sequence of runs, where we allow a run to be trivial i.e. be a single index.

Next, we show that within a run, we must have $s_1(i) < s_2(i)$ for all $i$. Let $S = \{l, l+1, \ldots l+k\}$ be a run. By optimality of $M_1, M_2$ at $\alpha^* - \epsilon$ and $\alpha^* + \epsilon$ respectively, we have:

$$\sum_{i \in S}\left(\frac{1}{\alpha^* - \epsilon}\lambda_i^{(1)} - (\alpha^* - \epsilon)\lambda_{s_1(i)}^{(2)}\right)^2 < \sum_{i \in S}\left(\frac{1}{\alpha^* - \epsilon}\lambda_i^{(1)} - (\alpha^* - \epsilon)\lambda_{s_2(i)}^{(2)}\right)^2$$

and

$$\sum_{i \in S}\left(\frac{1}{\alpha^* + \epsilon}\lambda_i^{(1)} - (\alpha^* + \epsilon)\lambda_{s_2(i)}^{(2)}\right)^2 < \sum_{i \in S}\left(\frac{1}{\alpha^* + \epsilon}\lambda_i^{(1)} - (\alpha + \epsilon)\lambda_{s_1(i)}^{(2)}\right)^2.$$

Respectively, these simplify to

$$-\sum_{i \in S}(\lambda_{s_1(i)}^{(2)} - \lambda_{s_2(i)}^{(2)})(-2\lambda_i^{(i)} + (\alpha^* - \epsilon)^2(\lambda_{s_1(i)}^{(2)} + \lambda_{s_2(i)}^{(2)})) > 0$$

and

$$\sum_{i \in S} \left( \lambda^{(2)}_{s_1(i)} - \lambda^{(2)}_{s_2(i)} \right) \left( -2\lambda^{(i)}_i + (\alpha^* + \epsilon)^2 \left( \lambda^{(2)}_{s_1(i)} + \lambda^{(2)}_{s_2(i)} \right) \right) > 0.$$

Summing these inequalities and cancelling $-2\lambda^{(i)}_i$, we have:

$$\sum_{i \in S} \left\{ (\alpha^* + \epsilon)^2 \left( \left( \lambda^{(2)}_{s_1(i)} \right)^2 + \left( \lambda^{(2)}_{s_2(i)} \right)^2 \right) - (\alpha^* - \epsilon)^2 \left( \left( \lambda^{(2)}_{s_1(i)} \right)^2 + \left( \lambda^{(2)}_{s_2(i)} \right)^2 \right) \right\} > 0.$$

Summing and reducing gives us

$$4\alpha^* \epsilon \left( \sum_{i \in S} \left( \lambda^{(2)}_{s_1(i)} \right)^2 - \sum_{i \in S} \left( \lambda^{(2)}_{s_2(i)} \right)^2 \right) > 0 \quad \text{and so} \quad \sum_{i \in S} \left( \lambda^{(2)}_{s_1(i)} \right)^2 > \sum_{i \in S} \left( \lambda^{(2)}_{s_2(i)} \right)^2.$$

However, since the $\lambda^{(2)}_j$ are monotonic nondecreasing, this means we cannot also have $s_1(i) > s_2(i)$ for all $i \in S$, since that would imply

$$\sum_{i=1}^{n_1} \left( \lambda^{(2)}_{s_1(i)} \right)^2 < \sum_{i=1}^{n_1} \left( \lambda^{(2)}_{s_2(i)} \right)^2.$$

Therefore, in a run of arbitrary length, all indices must move 'forward' (meaning that $s_1(i) < s_2(i)$ for all $i$ in the run), and so (since any pair of matchings optimal at such $\alpha$ define a set of runs) we must have $s_1(i) \leq s_2(i)$. This completes the proof of the lemma.

Thus, for three matchings $M_1$, $M_2$, $M_3$ which optimal at a sequence of $\alpha_1 \leq \alpha_2 \leq \alpha_3$, we must have $s_1(i) \leq s_2(i) \leq s_3(i)$ for all $i$. In particular, if $s_1(i) = s_3(i)$, we must also have $s_1(i) = s_2(i) = s_3(i)$.

**Theorem 12**. *If two matchings $M_1$ and $M_3$ yield optimal upper bounds for the linear distance $\tilde{D}^2$ at $\alpha_1$ and $\alpha_3$ respectively, and $f_{M_1}(\alpha_2) = f_{M_2}(\alpha_2)$ for some $\alpha_2$ s.t. $\alpha_1 \leq \alpha_2 \leq \alpha_3$, then either (1) $M_1$ and $M_3$ are optimal over the entire interval $[\alpha_1, \alpha_3]$ or (1) some other matching $M_2$ improves over $M_1$ and $M_3$ at $\alpha_2$.*

*Proof.* This follows directly from the facts that $f_{M_1}(\alpha)$ and $f_{M_2}(\alpha)$ (as defined in Eq (53)), can only meet at one real positive value of $\alpha$ (Lemma 8). Say that the cost curves for $M_1$ (known to be optimal at $\alpha = \alpha_1$) and $M_3$ (optimal at $\alpha = \alpha_3$) meet at $\alpha = \alpha_2$, and furthermore assume that $\alpha_1 \leq \alpha_2 \leq \alpha_3$. If some other matching $M_2$ improves over (meaning, has lesser obj. function value as a function of $\alpha$) $M_1$ or $M_3$ anywhere in the interval $[\alpha_1, \alpha_3]$, it must improve over both at $\alpha = \alpha_2$, since it may intersect each of these cost curves at most once on this interval. If $M_1$ and $M_3$ are both optimal at their intersection point (meaning no such distinct $M_2$ exists) then we know that no other matching improves on either of them over the the interval $[\alpha_1, \alpha_3]$ and may therefore mark it as explored during the outermost loop (otimization over $\alpha$) of Algorithm 1.

Together, the preceeding properties verify that our algorithm will indeed find the joint optimum over all $\alpha$ and $P$ (for fixed $t = c$, for $\tilde{D}$, subject to subpermutation constraints on $P$): it allows us to find the entire set of $P$ subpermutation matrices which appear on the lower convex hull of distance as a function of alpha.

## 4.3 Implementation details

We implement Algorithms 1 and 2 in the programming language "Python" (version 3.6.1) [34]. Numerical arrays were stored using the *numpy* package [35]. Our inner LAP solver was the package *lapsolver* [36]. Univariate optimization over $t$ and $\alpha$ was performed with the

'bounded' method of the *scipy.optimize* package [37], with bounds set at [0, 10.0] for each variable and a input tolerance of $10^{-12}$. Laplacians were computed with the *laplacian* method from the package *networkX* [38], and their eigenvalues were computed with *scipy.linalg.eigh*.

Because of numerical precision issues arising during eigenvalue computation, it can be difficult to determine when two matchings agree, using eigenvalue comparison. In practice we ignore this issue and assume that two matchings are only identical if they associate the same indices of the two lists of eigenvalues. This means we may be accumulating multiple equivalent representations of the same matching (up to multiplicity of eigenvalues) during our sweeps through $t$ and $\alpha$. We leave mitigating this inefficiency for future work.

Code for computing diffusion distance, both with our algorithm and with naive univariate optimiztion, may be found in the S1 Data associated with this paper, as well as a maintained GitHub repository [39].

## 5 Numerical experiments

### 5.1 Graph lineages

In this subsection we introduce several graph lineages for which we will compute various intra- and inter-lineage distances. Three of these are well-known lineages of graphs, and the fourth is defined in terms of a product of complete graphs:

*Path Graphs (Pa$_n$)*: 1D grid graphs of length $n$, with aperiodic boundary conditions.

*Cycle Graphs (Cy$_n$)*: 1D grid graphs of length $n$, with periodic boundary conditions.

*Square Grid Graphs (Sq$_n$)*: 2D grid graphs of dimensions $n$, with aperiodic boundary conditions. Sq$_n$ = Pa$_n$□Pa$_n$

*"Multi-Barbell" Graphs (Ba$_n$)*: Constructed as Cy$_n$□K$_n$, where $K_n$ is the complete graph on $n$ vertices.

These families are all illustrated in Fig 5.

Additionally, some examples distances between elements of these graph lineages are illustrated in Fig 6. In these tables we see that in general intra-lineage distances are small, and inter-lineage distances are large.

### 5.2 Numerical optimization methods

We briefly discuss here the other numerical methods we have used to calculate $\tilde{D}^2$ and $D^2$.

*Nelder-Mead in Mathematica* For very small graph pairs ($n_1 \times n_2 \leq 100$) we are able to find optimal $P$, $\alpha$, $t$ using constrained optimization in Mathematica 11.3 [40] using NMinimize, which uses Nelder-Mead as its backend by default. The size limitation made this approach unusable for any real experiments.

*Orthogonally Constrained Opt*. We also tried a variety of codes specialized for numeric optimization subject to orthogonality constraints. These included (1) the python package PyManopt [41], a code designed for manifold-constrained optimization; (2) gradient descent in Tensorflow using the penalty function $g(P) = c||P^T P - I||_F$ (with $c \ll 1$ a small positive constant weight) to maintain orthogonality, as well as (3) an implementation of the Cayley reparametrization method from [42] (written by the authors of that same paper). In our experience, these codes were slower, with poorer scaling with problem size, than combinatorial optimization over subpermutation matrices, and did not produce improved results on our optimization problem.

*Black-Box Optimization Over $\alpha$*. We compare in more detail two methods of joint optimization over $\alpha$ and $P$ when $P$ is constrained to be a subpermutation matrix in the diagonal basis for $L(G_1)$ and $L(G_2)$. Specifically, we compare our approach given in Algorithm 1 to univariate

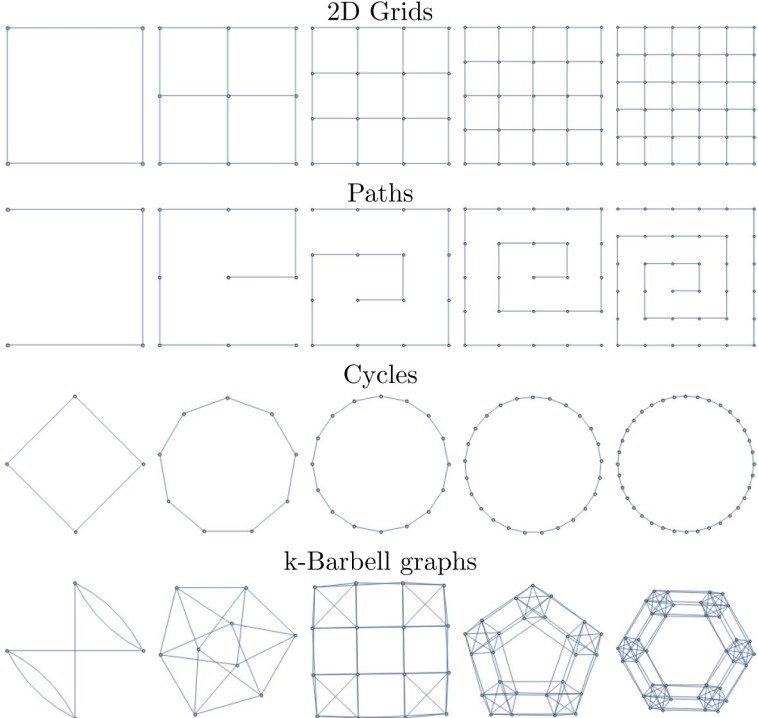

**Fig 5. Graph lineages used in multiple numerical experiments in the main text.**

optimization over $\alpha$, where each function evaluation consists of full optimization over $P$. Fig 7 shows the results of this experiment. We randomly sample pairs of graphs as follows:

1. $n_1$ is drawn uniformly from [5, 120].

2. $n_2$ is drawn uniformly from [$n_1$, $n_1$ + 60].

3. $G_1$ and $G_2$ are generated by adding edges according to a Bernoulli distribution with probability $p$. We ran 60 trials for each $p$ in {.125, .25, .375, .5, .625, .75, .875 }.

We compute the linear version of distance for each pair. Because our algorithm finds all of the local minima as a function of alpha, we compute the cost of the golden section approach as the summed cost of multiple golden section searches in alpha: one GS search starting from the initial bracket [$0.618\alpha^*$, $1.618\alpha^*$] for each local minimum $\alpha^*$ found by our algorithm. We see that our algorithm is always faster by at least a factor of 10, and occasionally faster by as much as a factor of $10^3$. This can be attributed to the fact that the golden section search is unaware of the structure of the linear assignment problem: it must solve a full $n_2 \times n_2$ linear assignment problem for each value of $\alpha$ it explores. In contrast, our algorithm is able to use information from prior calls to the LAP solver, and therefore solves a series of LAP problems whose sizes are monotonically nonincreasing.

## 5.3 Experiments

### 5.3.1 Triangle inequality violation of $D$ (exponential distance) and $\tilde{D}$ (linear distance).
As stated in Section 2.3, our full graph dissimilarity measure does not necessarily obey the triangle inequality. In this section we systematically explore conditions under which

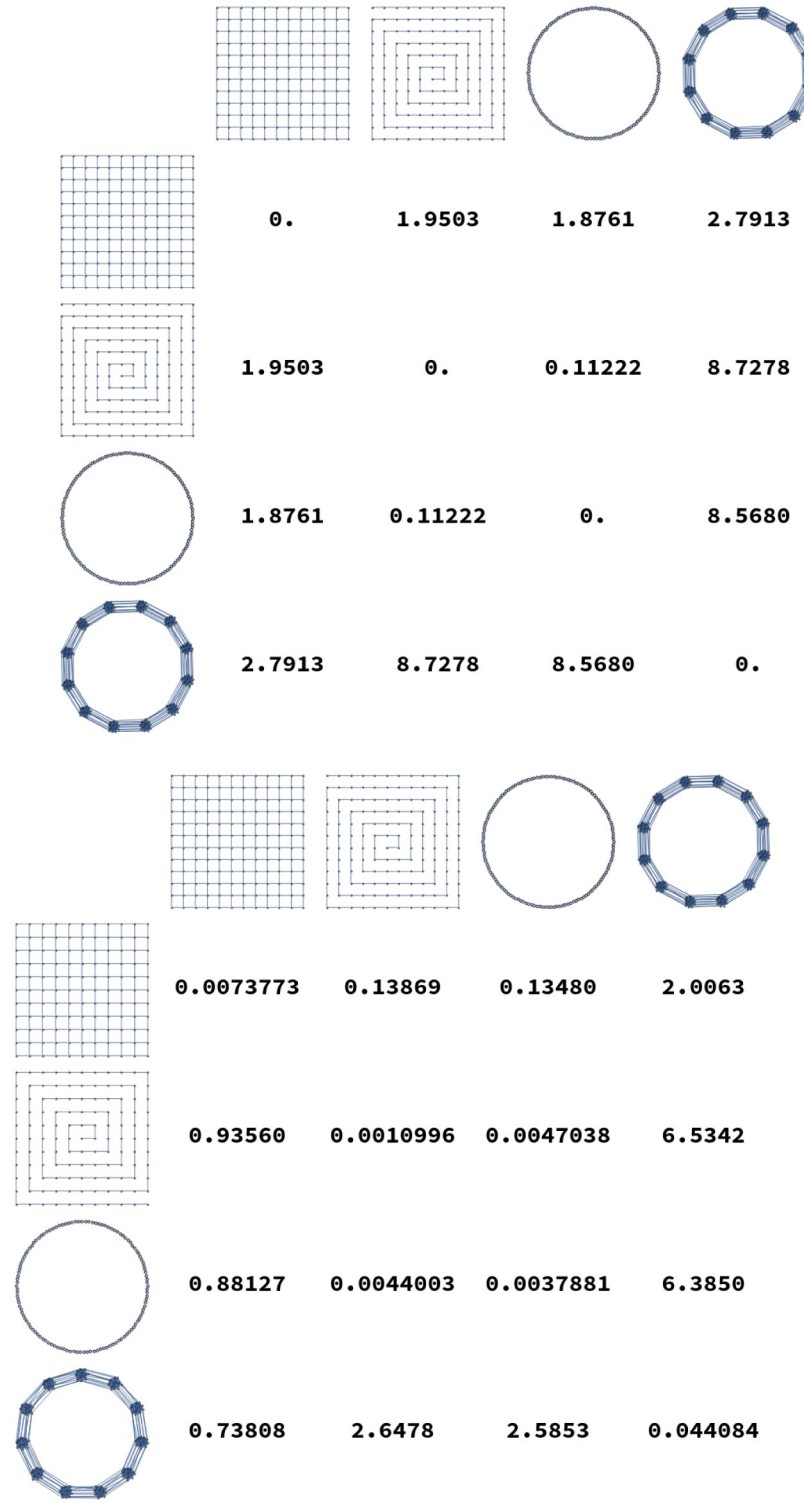

**Fig 6. Distances $D^2(G, H)$ calculated for several pairs of graphs.** The top plot shows distances where $G$ and $H$ are both chosen from {Grid$_{13 \times 13}$, $P_{169}$, $C_{169}$, Ba$_{13}$}. At bottom, distances are calculated from $G$ chosen in {Grid$_{12 \times 12}$, $P_{144}$, $C_{144}$, Ba$_{12}$} to $H$ chosen in {Grid$_{13 \times 13}$, $P_{169}$, $C_{169}$, Ba$_{13}$}. As expected, diagonal entries are smallest.

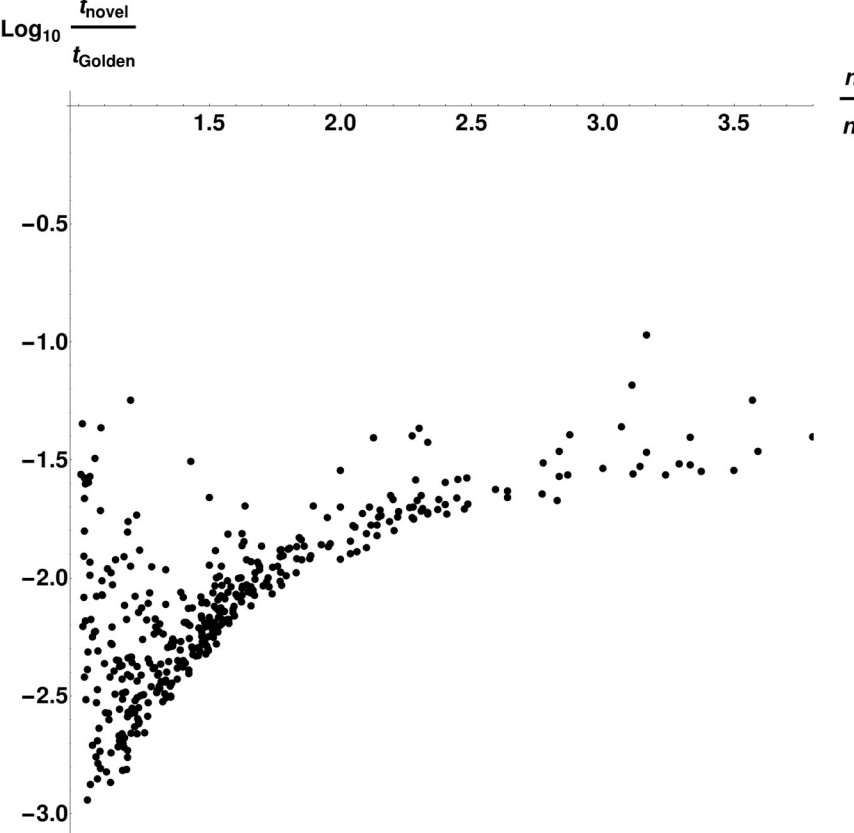

**Fig 7. Comparison of runtimes for our algorithm and bounded golden section search over the same interval [$10^{-6}$, 10].** Runtimes were measured by a weighted count of evaluations of the Linear Assignment Problem solver, with an $n \times n$ linear assignment problem counted as $n^3$ units of cost. Because our algorithm recovers the entire lower convex hull of the objective function as a function of $\alpha$, we compute the cost of the golden section search as the summed cost of multiple searches, starting from an interval bracketing each local optimum found by our algorithm. We see that our algorithm is much less computationally expensive, sometimes by a factor of $10^3$. The most dramatic speedup occurs in the regime where $n_1 \ll n_2$. Graphs were generated by drawing $n_1$ uniformly from [5, 120], drawing $n_2$ uniformly from [$n_1$, $n_1 + 60$], and then adding edges according to a Bernoulli distribution with $p$ in {.125, .25, .375, .5, .625, .75, .875 } (60 trials each).

the triangle inequality is satisfied or not satisfied. We generate triplets $G_1$, $G_2$, $G_3$ of random graphs of sizes $n_i$ for $n_1 \in [5, 30]$, $n_2 \in [n_1, n_1 + 30]$, and $n_3 \in [n_2, n_2 + 30]$ by drawing edges from the Bernoulli distribution with probability $p$ (we perform 4500 trials for each $p$ value in [.125, .25, .375, .5, .625, .75, .875]). We compute the distance $\tilde{D}(G_i, G_k)$ (for $(i, k) \in \{(1, 3), (1, 2), (2, 3)\}$). The results may be seen in Fig 8. In this figure we plot a histogram of the "discrepancy score"

$$\text{Disc}(G_1, G_2, G_3) = \tilde{D}(G_1, G_3) / (\tilde{D}(G_1, G_2) + \tilde{D}(G_2, G_3)), \quad (60)$$

which measures the degree to which a triplet of graphs violates the triangle inequality (i.e. falls outside of the unit interval [0, 1]), for approximately $3 \times 10^4$ such triplets. It is clear that, especially for the linear definition of the distance, the triangle inequality is not always satisfied. However, we also observe that (for graphs of these sizes) the discrepancy score is bounded: no triple violates the triangle inequality by more than a factor of approximately 1.8. This is shown

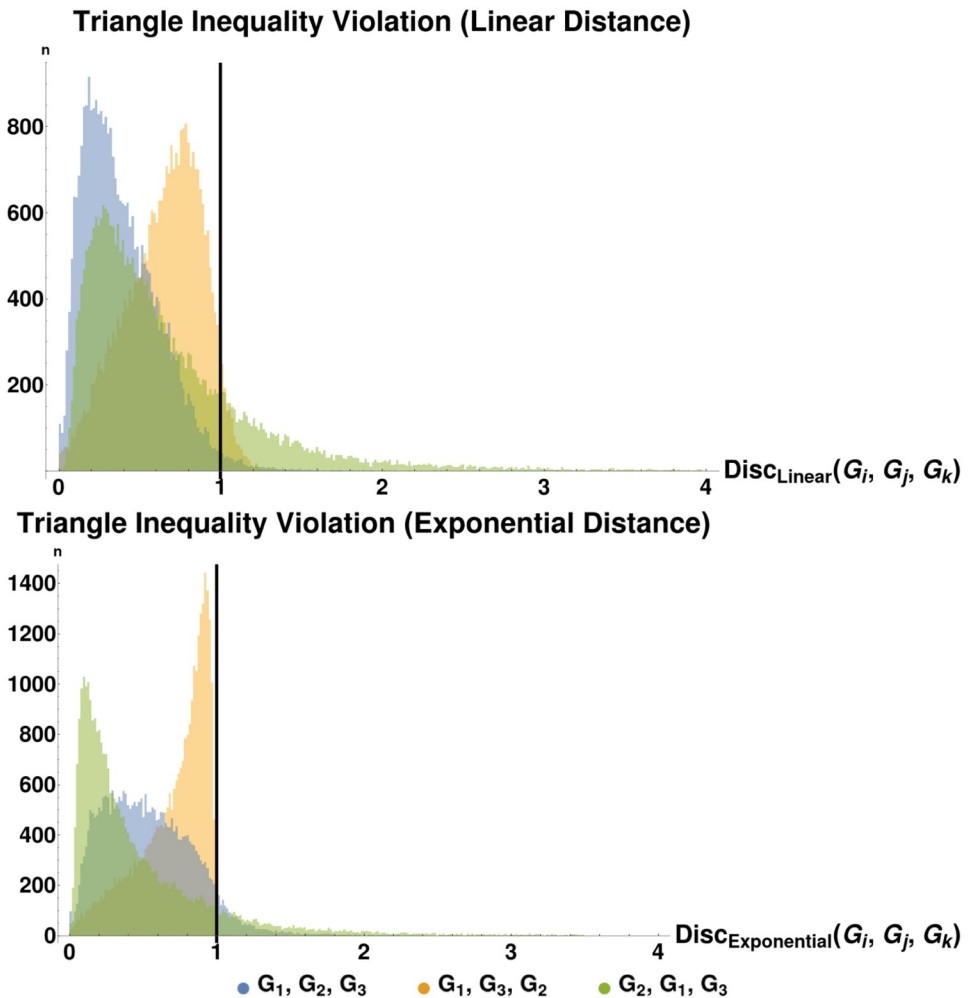

**Fig 8. Histograms of triangle inequality violation.** These plots show the distribution of $\mathrm{Disc}(G_1, G_2, G_3)$, as defined in the text, for the cases (a) top: the linear or small-time version of distance and (b) bottom: the exponential or arbitrary-time version of distance. We see that for the sizes of graph we consider, the largest violation of the triangle inequality is bounded, suggesting that our distance measure may be an infra-$\rho$-pseudometric for some value of $\rho \approx 1.8$ (linear version) or $\rho \approx 5.0$ (exponential version). See Table 1 for a summary of the distance metric variants introduced in this paper. We also plot the same histogram for out-of-order (by vertex size) graph sequences: $\mathrm{Disc}(G_2, G_1, G_3)$ and $\mathrm{Disc}(G_3, G_2, G_1)$. Each plot has a line at $x = 1$, the maximum discrepancy score for which the underlying distances satisfy the triangle inequality.

by the histogram of discrepancies in Fig 8. Additionally, the triangle inequality is satisfied in 28184 (95.2%) of cases.

We see similar but even stronger results when we run the same experiment with $D^2$ instead of $\tilde{D}^2$; these may also be seen in Fig 8. We calculated the discrepancy score analogously, but with $D$ substituted for $\tilde{D}$. We see similarly that the degree of violation is bounded. In this case, no triple violated the triangle inequality by a factor of more than 5, and in this case the triangle inequality was satisfied in 99.8% of the triples. In both of these cases, the triangle inequality violations may be a result of our optimization procedure finding local minima/maxima for one or more of the three distances computed. We also repeat the above procedure for the same triplets of graphs, but with distances computed not in order of increasing vertex size: calculating $\mathrm{Disc}(G_2, G_1, G_3)$ and $\mathrm{Disc}(G_3, G_2, G_1)$. All of these results are plotted in Fig 8.

**Table 2. Mean distances between graphs in several lineages.** For two lineages $G_1, G_2$... (listed at left) and $H_1, H_2, \ldots$ (listed at the top), each entry shows the mean distance $D(G_i, H_{i+1})$ (where the average is taken over $i = 1$ to 12). As expected, we see that the distance from elements of a graph lineage to other members of the same lineage (the diagonal entries of the table) is smaller than distances taken between lineages. Furthermore as expected, 1D paths are more similar (but not equal) to 1D cycles than to other graph lineages.

|  | Square Grids | Paths | Cycles | Multi-Barbells |
|---|---|---|---|---|
| Square Grids | 0.0096700 | 0.048162 | 0.046841 | 0.63429 |
| Paths | 0.30256 | 0.0018735 | 0.010300 | 2.1483 |
| Cycles | 0.27150 | 0.0083606 | 0.0060738 | 2.0357 |
| Multi-Barbells | 0.21666 | 0.75212 | 0.72697 | 0.029317 |

**5.3.2 Intra- and inter-lineage distances.** We compute pairwise distances for sequences of graphs in the graph lineages displayed in Fig 5. For each pair of graph families (Square Grids, Paths, Cycles, and Multi-Barbells), we compute the distance from the $i$th member of one lineage to the $(i + 1)$-st member of each other lineage, and take the average of the resulting distances from $i = 1$ to $i = 12$. These distances are listed in Table 2. As expected, average distances within a lineage are smaller than the distances from one lineage to another.

We note here that the idea of computing intra- and inter- lineage distances is similar to recent work [43] computing distances between *graph ensembles*: certain classes of similarly-generated random graphs. Graph diffusion distance has been previously shown (in [43]) to capture key structural information about graphs; for example, GDD is known to be sensitive to certain critical transitions in ensembles of random graphs as the random parameters are varied. This is also true for our time dilated version of GDD. More formally: let $G_p$ and $G'_p$ represent random graphs on $n$ vertices, drawn from the Erdős-Renyi distribution with edge probability $p$. Then $D(G_p, G'_p)$ has a local maximum at $p = \frac{1}{n}$, representing the transition between disconnected and connected graphs. This is true for our distance as well as the original version due to Hammond.

**5.3.3 Graph limits.** Here, we provide preliminary evidence that graph distance measures of this type may be used in the definition of a *graph limit*—a graphlike object which is the limit of an infinite sequence of graphs. This idea has been previously explored, most famously by Lovász [7], whose definition of a graph limit (called a *graphon*) is as follows: Recall the definition of graph cut-distance $D_{\text{cut}}(G, H)$ from Eq 3, namely: the cut distance is the maximum discrepancy in sizes of edge-cuts, taken over all possible subsets of vertices, between two graphs on the same vertex-set. A graphon is then an equivalence class of Cauchy sequences of graphs, under the equivalence relation that two sequences $G_1, G_2, \ldots$ and $H_1, H_2, \ldots$ are equivalent if $D_{\text{cut}}(G_i, H_i)$ approaches 0 as $n \to \infty$. Here we are calling a sequence of graphs "Cauchy" if for any $\epsilon > 0$ there is some $N$ such that for all $n, m \geq N$, $D_{\text{cut}}(G_n, G_m) < \epsilon$.

We propose a similar definition of graph limits, but with our diffusion distance substituted as the distance measure used in the definition of a Cauchy sequence of graphs. Hammond et. al. argue in [1] why their variant of diffusion distance may be a more descriptive distance measure than cut-distance. More specifically, they show that on some classes of graphs, some edge deletions 'matter' much more than others: removal of a single edge changes the diffusive properties of the graph significantly. However, the graph-cut distance between the new and old graphs is the same, regardless of which edge has been removed, while the diffusion distance captures this nuance. For graph limits, however, our generalization to *unequal-sized graphs* via $P$ is of course essential. Furthermore, previous work [44] on sparse graph limits has shown that in the framework of Lovász all sequences of sparse graphs converge (in the infinite-size limit) to the zero graphon. Graph convergence results specific to sparse graphs include the Benjamini-Schramm framework [45], in which graph sequences are compared using the

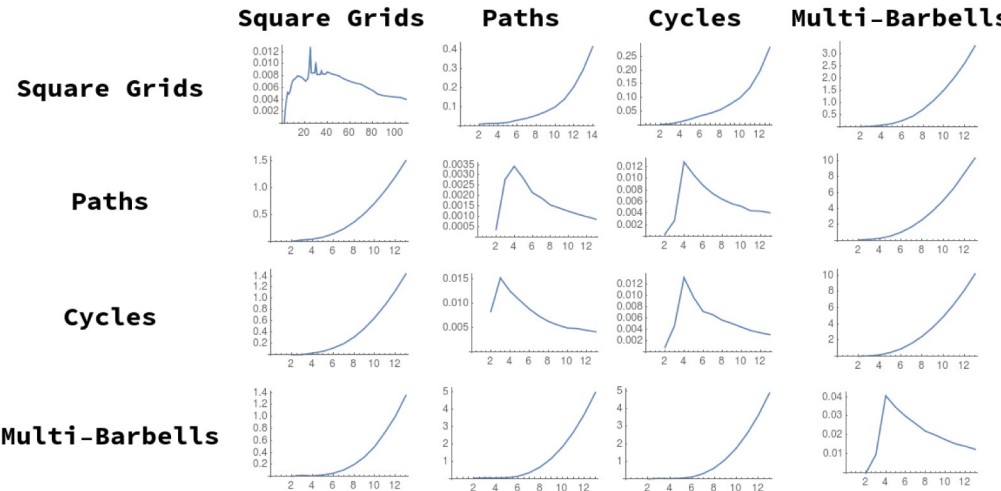

**Fig 9. Cauchy-like behavior of graph distance as a function of sequence index, *n*.** The distance between successive square grids and all other graph sequences appears to diverge (the same behavior is seen for k-barbells). Notably, the distance between Grid$_{n \times n}$ and Grid$_{(n+1) \times (n+1)}$ does not appear to converge, until much higher values of $n$ ($n > 100$) than the other convergent series. This may be because the distances calculated are an upper bound, and may be converging more slowly than the 'true' optima.

distributional limits of subgraph frequencies. These two graph comparison methods both have the characteristic that the "limit object" of a sequence of graphs is rigorously defined. In this section we attempt to show empirically that such a limit object of graph sequences under GDD may exist, and therefore merit further investigation.

We examine several sequences of graphs of increasing size for the required Cauchy behavior (in terms of our distance measure) to justify this variant definition of a "graph limit". For each of the graph sequences defined in Section 5.1, we examine the distance between successive members of the sequence, plotting $D^2(G_n, H_{n+1})$ for each choice of $G$ and $H$. These sequences of distances are plotted in Fig 9.

In this figure, we see that generally distance diverges between different graph lineages, and converges for successive members of the same lineage, as $n \to \infty$. We note the exceptions to this trend:

1. The distances between $n$-paths and $n + 1$-cycles appear to be converging; this is intuitive, as we would expect that difference between the two spectra due to distortion from the ends of the path graph would decrease in effect as $n \to \infty$.

2. We also show analytically, under similar assumtions, that the distance between successive path graphs also shrinks to zero (Theorem 14).

We do not show that all similarly-constructed graph sequences display this Cauchy-like behavior. We hope to address this deeper question, as well as a more formal exploration of the limit object, with one or more modified versions of the objective function (see Section 3.6).

**5.3.4 Limit of path graph distances.** In this section, we demonstrate analytically that the sequence of path graphs of increasing size is Cauchy in the sense described by the previous section. In the following theorem (Theorem 14), we assume that the optimal value of $t$ approaches some value $\tilde{t}$ as $n \to \infty$. We have not proven this to be the case, but have observed this behavior for both square grids and path graphs (see Fig 10 for an example of this behavior). Lemmas 13 and 14 show a related result for path graphs; we note that the spectrum of the Laplacian (as

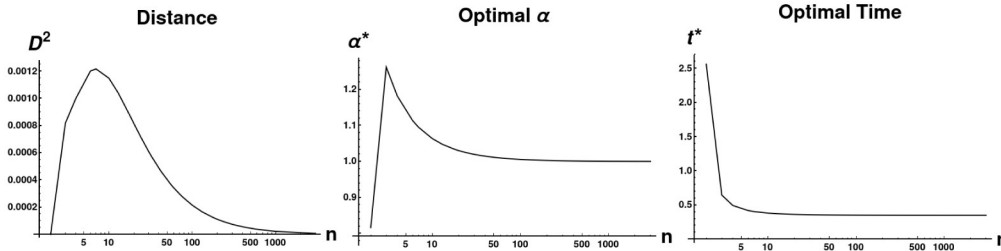

**Fig 10. Limiting behavior of *D* and two parameters as path graph size approaches infinity.** All distances were calculated between Path$_n$ and Path$_{n+1}$. We plot the value of the objective function, as well as the optimal values of $\alpha$ and $t$, as $n \to \infty$. Optimal $\alpha$ rapidly approach 1 and the optimal distance tends to 0. Additionally, the optimal $t$ value approaches a constant ($t \approx .316345$), providing experimental validation of the assumption we make in proving Theorem 14.

we define it in this paper) of a path graph of size $n$ is given by

$$\lambda_k = -2 + 2\cos\frac{k\pi}{n-1} \qquad k \in \{0...n-1\}.$$

**Lemma 13**. *For any finite k, t, we have*

$$\lim_{n\to\infty} n\left(e^{t\left(-2+2\cos\left(\frac{\pi k}{n}\right)\right)} - e^{t\left(-2+2\cos\left(\frac{\pi k}{n+1}\right)\right)}\right)^2 = 0$$

*Proof.* Clearly for finite $k, t$

$$\lim_{n\to\infty}\left(e^{t\left(-2+2\cos\left(\frac{\pi k}{n}\right)\right)} - e^{t\left(-2+2\cos\left(\frac{\pi k}{n+1}\right)\right)}\right) = 0$$

Then,

$$\lim_{n\to\infty} n\left(e^{-2+2\cos\left(\frac{\pi k}{n}\right)} - e^{-2+2\cos\left(\frac{\pi k}{n+1}\right)}\right)$$

$$= \lim_{n\to\infty} \frac{\left(e^{-2+2\cos\left(\frac{\pi k}{n}\right)} - e^{-2+2\cos\left(\frac{\pi k}{n+1}\right)}\right)}{\frac{1}{n}}$$

Evaluating this expression requires applying L'Hôpital's rule. Hence, we have:

$$\lim_{n\to\infty} \frac{\left(e^{-2+2\cos\left(\frac{\pi k}{n}\right)} - e^{-2+2\cos\left(\frac{\pi k}{n+1}\right)}\right)}{\frac{1}{n}}$$

$$= \lim_{n\to\infty} \frac{2\pi kt\left(\frac{\sin\left(\frac{\pi k}{n}\right)e^{2t\left(\cos\left(\frac{\pi k}{n}\right)-1\right)}}{n^2} - \frac{\sin\left(\frac{\pi k}{n+1}\right)e^{2t\left(\cos\left(\frac{\pi k}{n+1}\right)-1\right)}}{(n+1)^2}\right)}{\frac{-1}{n^2}}$$

$$= 2\pi kt \lim_{n\to\infty}\left(\frac{n^2\sin\left(\frac{\pi k}{n+1}\right)e^{2t\left(\cos\left(\frac{\pi k}{n+1}\right)-1\right)}}{(n+1)^2} - \sin\left(\frac{\pi k}{n}\right)e^{2t\left(\cos\left(\frac{\pi k}{n}\right)-1\right)}\right).$$

Since both of the limits

$$\lim_{n\to\infty} \left( \frac{n^2 \sin\left(\frac{\pi k}{n+1}\right) e^{2t\left(\cos\left(\frac{\pi k}{n+1}\right)-1\right)}}{(n+1)^2} \right)$$

and

$$\lim_{n\to\infty} \left( -\sin\left(\frac{\pi k}{n}\right) e^{2t\left(\cos\left(\frac{\pi k}{n}\right)-1\right)} \right)$$

exist (and are 0),

$$2\pi k t \lim_{n\to\infty} \left( \frac{n^2 \sin\left(\frac{\pi k}{n+1}\right) e^{2t\left(\cos\left(\frac{\pi k}{n+1}\right)-1\right)}}{(n+1)^2} - \sin\left(\frac{\pi k}{n}\right) e^{2t\left(\cos\left(\frac{\pi k}{n}\right)-1\right)} \right) = 0$$

and therefore

$$\lim_{n\to\infty} n \left( e^{t\left(-2+2\cos\left(\frac{\pi k}{n}\right)\right)} - e^{t\left(-2+2\cos\left(\frac{\pi k}{n+1}\right)\right)} \right)^2 = 0$$

**Theorem 14**. *If* $\lim_{n\to\infty} \arg\sup_t D^2\left(\mathrm{Pa}_n, \mathrm{Pa}_{n+1}|t\right)$ *exists, then*:

$$\lim_{n\to\infty} D^2(\mathrm{Pa}_n, \mathrm{Pa}_{n+1}) = 0.$$

*Proof.* Assume that $\lim_{n\to\infty} \arg\sup_t D^2\left(\mathrm{Pa}_n, \mathrm{Pa}_{n+1}|t\right) = \tilde{t}$. Then, we must have

$$\lim_{n\to\infty} D^2(\mathrm{Pa}_n, \mathrm{Pa}_{n+1}) \quad \leq \lim_{n\to\infty} D^2(\mathrm{Pa}_n, \mathrm{Pa}_{n+1}|\tilde{t})$$

Hence, it remains only to prove that

$$\lim_{n\to\infty} D^2(\mathrm{Pa}_n, \mathrm{Pa}_{n+1}|t) = 0$$

for any finite $t$ (which will then include $\tilde{t}$). First, for any particular $(n + 1) \times n$ subpermutation matrix $S$, note that

$$D^2(\mathrm{Pa}_n, \mathrm{Pa}_{n+1}|t) \quad = \inf_{\alpha>0} \inf_{P|\mathcal{C}(P)} D^2(\mathrm{Pa}_n, \mathrm{Pa}_{n+1}|t, P, \alpha)$$

$$\leq D^2(\mathrm{Pa}_n, \mathrm{Pa}_{n+1}|t, \alpha = 1, U_{n+1}^T S U_n)$$

Here, $U_n$ and $U_{n+1}$ are the matrices which diagonalize $L(\mathrm{Pa}_n)$ and $L(\mathrm{Pa}_{n+1})$ respectively (note also that a diagonalizer of a matrix $L$ also diagonalizes $e^L$). If at each $n$ we select $S$ to be the subpermutation $S = \begin{bmatrix} I \\ 0 \end{bmatrix}$, then (using the same argument as in Theorem 7) the objective

function simplifies to:

$$D^2\left(\mathrm{Pa}_n, \mathrm{Pa}_{n+1} | t, P = U_{n+1}^T S U_n, \alpha = 1\right)$$

$$= \left\| S e^{c\Lambda_{\mathrm{Pa}_n}} - e^{c\Lambda_{\mathrm{Pa}_{n+1}}} S \right\|_F^2$$

$$= \sum_{k=0}^{n-1} \left( e^{c\left(-2+2\cos\left(\frac{\pi k}{n}\right)\right)} - e^{c\left(-2+2\cos\left(\frac{\pi k}{n+1}\right)\right)} \right)^2$$

$$\leq \max_{0 \leq k \leq n-1} n \left( e^{c\left(-2+2\cos\left(\frac{\pi k}{n}\right)\right)} - e^{c\left(-2+2\cos\left(\frac{\pi k}{n+1}\right)\right)} \right)^2$$

By Lemma 13, for any finite $k$, $t$, we have

$$\lim_{n \to \infty} n \left( e^{t\left(-2+2\cos\left(\frac{\pi k}{n}\right)\right)} - e^{t\left(-2+2\cos\left(\frac{\pi k}{n+1}\right)\right)} \right)^2 = 0$$

So for any $\epsilon > 0$, $\exists N$ such that when $n \geq N$, for any $c$, $k$,

$$n \left( e^{c\left(-2+2\cos\left(\frac{\pi k}{n}\right)\right)} - e^{c\left(-2+2\cos\left(\frac{\pi k}{n+1}\right)\right)} \right)^2 < \epsilon$$

But then

$$\sum_{k=0}^{n-1} \left( e^{c\left(-2+2\cos\left(\frac{\pi k}{n}\right)\right)} - e^{c\left(-2+2\cos\left(\frac{\pi k}{n+1}\right)\right)} \right)^2 < \epsilon$$

as required. Thus, the Cauchy condition is satisfied for the lineage of path graphs $\mathrm{Pa}_n$

Given a graph lineage which consists of levelwise box products between two lineages, it seems natural to use our upper bound on successive distances between graph box products to prove convergence of the sequence of products. As an example, the lineage consisting of square grids is the levelwise box product of the lineage of path graphs with itself. However, in this we see that this bound may not be very tight. Applying Eq (41) from Theorem 5, we have (for any $t_c$, $\alpha_c$):

$$
\begin{aligned}
D(\mathrm{Sq}_n, \mathrm{Sq}_{n+1}) \quad &\leq D(\mathrm{Sq}_n, \mathrm{Sq}_{n+1} | t_c, \alpha_c) \\
&\leq D\left(\mathrm{Pa}_{n+1}, \mathrm{Pa}_{n+1} | t_c, \alpha_c\right) \left( \left\| e^{\frac{t_c}{a_c} L(\mathrm{Pa}_n)} \right\|_F \right. \\
&\qquad \left. + \left\| e^{t_c a_c L(\mathrm{Pa}_{n+1})} \right\|_F \right)
\end{aligned}
$$

As we can see in Fig 11, the right side of this inequality seems to be tending to a nonzero value as $n \to \infty$, whereas the actual distance (calculated by our optimization procedure) appears to be tending to zero.

**5.3.5 Shape analysis for discretized meshes.** In this section we demonstrate that graph diffusion distance captures structural properties of 3D point clouds. Ten 3D meshes (see Fig 12 for an illustration of the meshes used) were chosen to represent an array of objects with varying structural and topological properties. Not all of the mesh files chosen are simple manifolds: for example, the "y-tube" is an open-ended cylinder with a fin around its equator. Each mesh was used to produce multiple graphs, via the following procedure:

1. Subsampling the mesh to 1000 points;

2. Performing a clustering step on the new point cloud to identify 256 cluster centers;

3. Connecting each cluster center to its 16 nearest neighbors in the set of cluster centers.

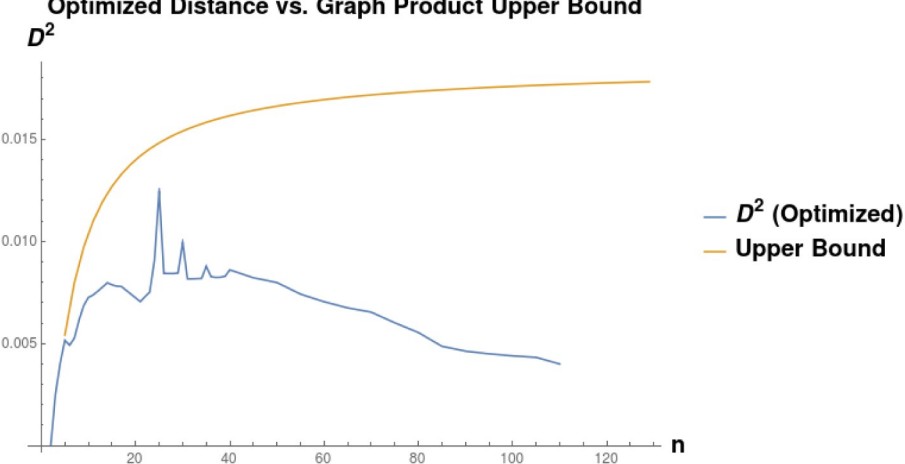

**Fig 11. Comparison of the distance $D(\mathrm{Sq}_n, \mathrm{Sq}_{n+1})$ as a function of $n$, to the upper bound calculated as the optimum of distance between $\mathrm{Pa}_n$ and $\mathrm{Pa}_{n+1}$.** We see that the upper found converges to some constant $D \approx 0.01782$, whereas the actual distance appears to be converging to 0 as $n \to \infty$.

Since each pass of this procedure (with different random seeds) varied in Step 1, each pass produced a different graph. We generated 20 graphs for each mesh, and compared the graphs using GDD.

The results of this experiment can be seen in Fig 13. This Figure shows the three first principal components of the distance matrix of GDD on the dataset of graphs produced as described above. Each point represents one graph in the dataset, and is colored according to the mesh which was used to generate it. Most notably, all the clusters are tight and do not overlap. Close clusters represent structurally similar objects: for example, the cluster of graphs from the tube mesh is very close to the cluster derived from the tube with an equatorial fin. This synthetic dataset example demonstrates that graph diffusion distance is able to compare structural information about point clouds and meshes.

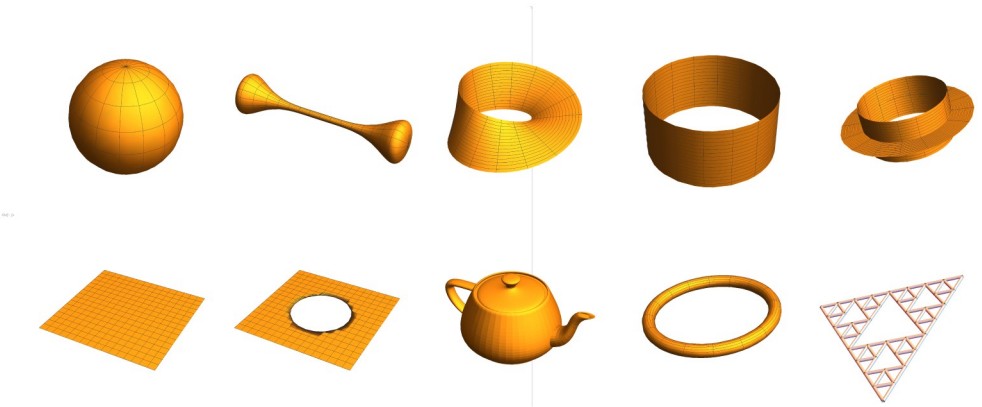

**Fig 12. 3D meshes used in the shape analysis experiment.** Each mesh was used to produce several sampled discretizations, which were then compared using GDD.

## Embedding of Graph Diffusion Distances in $\mathbb{R}^3$

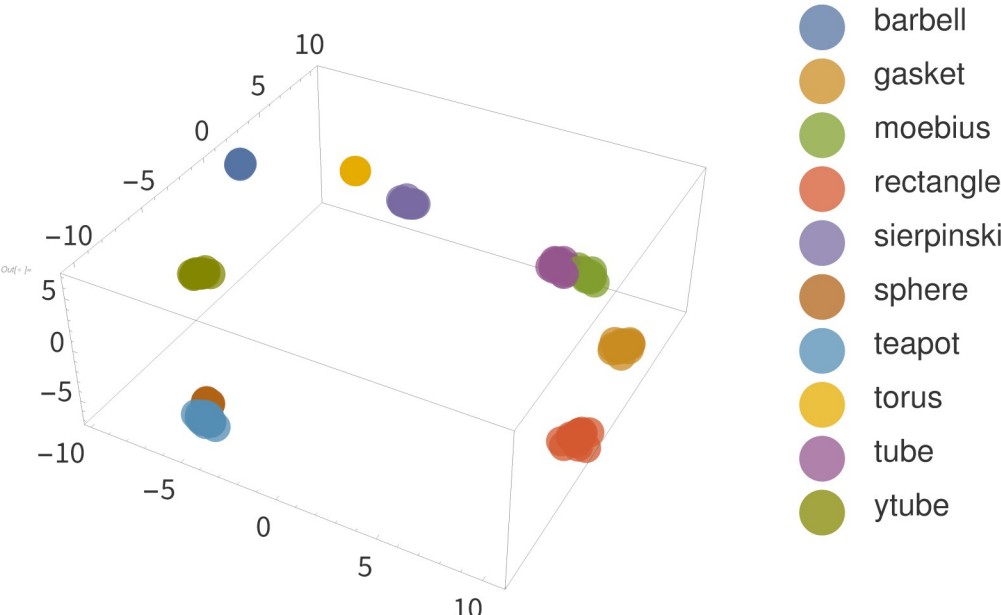

**Fig 13. Embedding of pairwise distances between mesh discretizations.** We see that GDD clusters each category of mesh tightly, and furthermore that clusters are nearby when they are structurally similar meshes, and distant otherwise. Axes represent the three principal components of the distance matrix and are thus unitless.

# 6 Applications and future work

We briefly discuss possible applications of both our distance metric and our procedure for calculating the relevant minima.

## 6.1 Algebraic multigrid

The need for prolongation / restriction operators arises naturally in the Algebraic MultiGrid (AMG) context, where a hierarchy of progressively coarser meshes are constructed, with the goal of speeding convergence of a model with local update ("smoothing") rules. A model with modes of behavior at wavelengths which are much larger than the neighborhood of one update will take many update steps to converge. Thus, the goal in AMG is to iteratively construct a series of coarsened meshes, so that update steps at the coarser scales can address coarser modes of behavior. A fine-scale model state is translated into a coarse-scale state via a "restriction" operator. After a coarse-scale smoothing step, the new coarse state is translated back to the fine-scale by "prolonging" it. Our procedure for calculating $P$ could be incorporated as a preprocessing step, in the case where the series of meshes are known in advance; otherwise, the $P$ from the previous round of coarsening could be used as the initial conditions to a modified version of our solver. In either case, the matrix $P$ is a natrual choice of prolongation/ restriction operator for this type of coarsening scheme, since it optimally transforms the Laplacian of one graph into another.

## 6.2 Graph limits

In this work we briefly introduce a new definition of graph limits based on the diffusion distance, which raises several natural questions: What does the "limit" of a sequence of graphs under diffusion distance look like? Are there pairs of sequences that converge to the same such object, as in the example of path graphs and cycle graphs? Can we separate graph sequences into equivalence classes based on which of these they converge to? We hope to address these questions in future work.

## 6.3 Graph convolutional networks

Graph convolutional networks (GCNs) are a variant of the *convolutional neural networks* (CNNs) widely used in machine vision. In the same way that CNNs learn a set of trained image filters and apply them across multiple spatial locations in an image, GCNs learn a set of filters which are applied to local neighborhoods of a graph. One implementation of GCNs due to Kipf and Welling [46] uses a Chebyshev polynomial of the Laplacian matrix as an approximation of the graph fourier transform, demonstrating comparable results to the full transform but far fewer multiplication operation needed. However, construction of pooling operators for GCNs is still an area of open research. Since our $P$ is a restriction operator that preserves information about the Laplacian, it is natural to use it as a pooling operator in this type of model. In [47] we make use of the optimization procedure described in this manuscript to find optimal (in the sense of Graph Diffusion Distance) coarsenings of a graph representing a protein nanotube. Since our procedure yields a $P$ matrix which maps between the coarse and fine graphs, we are then able to define a machine learning model which learns to reproduce the protein's energetic dynamics at multiple spatial scales. As a result of this multiscale construction, this multiscale model is more computationally efficient by an order of magnitude.

## 6.4 Graph clustering

We can also use the diffusion distance and its variants to compare graphs or neighborhoods of graphs for structural similarity, independent of graph size. Section 5.3.5 showcases an example of doing this with a small synthetic database of 3D point clouds derived from mesh files of several well-known objects. This is similar to the approach of [4] for comparing point clouds in 2D and 3D, in the sense that both approaches optimize an objective function based to a matching between elements of the two graphs. This type of similarity measure may then be used to convert a dataset of graphs to a distance-to-cluster-centers representation, or for any other of the typical methods used in machine learning for converting sets of pairwise distances into fixed-length feature vectors ($k$-medoids, kernel methods, multidimensional scaling, etc.). In this setting, our distance measure has an additional benefit: since computing it yields an explicit projection operator between the nodes of the graphs, we may use the set of $P$ we compute to project signals (e.g. labels on the vertices of each graph in the dataset) to a common space.

## 7 Conclusion

In this work, we present a novel generalization of graph diffusion distance which allows for comparison of graphs of inequal size. We consider several variants of this distance measure to account for sparse maps between the two graphs, and for maps between the two graphs which are optimal given a fixed time-dilation factor $\alpha$. We prove several important theory properties of distances in this family of measures, including triangle inequalities in some cases and Cauchy-like behavior of some graph sequences. We present a new procedure for optimizing the

objective function defined by our distance measure, prove the correctness of this procedure, and demonstrate its efficiency in comparison to univariate search over the dilation parameter, $\alpha$. Numerical experiments suggest that this dissimilarity score satisfies the triangle inequality up to some constant $\rho \approx 2.1$. We demonstrate that this measure of graph distance may be used to compare graph lineages (families of exponentially-growing graphs with shared structure), and additionally that certain lineages display Cauchy-sequence like behavior as the graph size approaches infinity. We suggest several possible applications of our distance measure to scientific problems in the contexts of pattern matching and machine learning.

## Supporting information

**S1 Data.**
(ZIP)

## Acknowledgments

The authors would like to gratefully acknowledge the hospitality of the Sainsbury Laboratory (Cambridge University, Cambridge, UK) and the Center for NonLinear Studies (Los Alamos National Laboratory, Los Alamos, NM, US).

## Author Contributions

**Conceptualization:** C. B. Scott, Eric Mjolsness.

**Formal analysis:** C. B. Scott, Eric Mjolsness.

**Funding acquisition:** C. B. Scott, Eric Mjolsness.

**Investigation:** C. B. Scott, Eric Mjolsness.

**Methodology:** C. B. Scott.

**Project administration:** Eric Mjolsness.

**Resources:** C. B. Scott, Eric Mjolsness.

**Software:** C. B. Scott.

**Validation:** C. B. Scott.

**Visualization:** C. B. Scott.

**Writing – original draft:** C. B. Scott, Eric Mjolsness.

**Writing – review & editing:** C. B. Scott, Eric Mjolsness.

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
