## [Decision Letter · Decision Letter 0]

23 Nov 2020

PONE-D-20-13130

Graph diffusion distance: Properties and efficient

computation

PLOS ONE

Dear Dr. Scott,

Thank you for submitting your manuscript to PLOS ONE. After careful consideration, we feel that it has merit but does not fully meet PLOS ONE’s publication criteria as it currently stands. Therefore, we invite you to submit a revised version of the manuscript that addresses the points raised during the review process.

We look forward to receiving your revised manuscript.

Kind regards,

Gabriele Oliva, Ph.D

Academic Editor

PLOS ONE

Journal Requirements:

2. Please ensure that you refer to Figure 5 in your text as, if accepted, production will need this reference to link the reader to the figure.

Additional Editor Comments:

Two reviews were collected. Reviewer 1 suggests a minor revision and provides useful hints for improvement. Reviewer 2 is more critical, especially regarding the significance of the results. After carefully reviewing the paper myself, I believe the paper can be publishable after a minor revision.

Reviewers' comments:

Reviewer's Responses to Questions

**Comments to the Author**

1. Is the manuscript technically sound, and do the data support the conclusions?

Reviewer #1: Yes

Reviewer #2: Yes

2. Has the statistical analysis been performed appropriately and rigorously? 

Reviewer #1: Yes

Reviewer #2: N/A

3. Have the authors made all data underlying the findings in their manuscript fully available?

Reviewer #1: No

Reviewer #2: Yes

4. Is the manuscript presented in an intelligible fashion and written in standard English?

Reviewer #1: Yes

Reviewer #2: Yes

5. Review Comments to the Author

Reviewer #1: This article is great. I have been working in the graph distance world for several years now, and I found this contribution to be quite thorough. Whenever I had a "but what about..." question, the authors seemed to answer it in the text right away. As with all graph distance papers, however, there are always more questions to be asked, and I do have a few that I would like the authors to address (see attached PDF). Apart from these minor notes, I would like to see this work published in PLOS ONE.

Reviewer #2: I recommend a major revision to address certain concerns about the significance of the results. In particular, it seems that most of the results study how to compute upper bounds on the proposed dissimilarity measures, but I would like to see more explicitly worked out applications of the measures, so that the reader can see why

they are useful.

1.) The authors may wish to cite some relevant works on the use of optimal transport to define distances between graphs. See, e.g., GOT: An Optimal Transport Framework for Graph Comparison, by Maretic et al., NeurIPS 2019.

2.) The paper mentions a few possible applications. I think that it is important that at least one of these applications be fleshed out (e.g., with experiments) so that we can see the practical utility of the new distance measure. For instance, the application to graph convolutional networks would be good to expand with some experiments.

3.) Regarding the graph limit application, it seems to me that one of the useful aspects of cut distance in defining graph limits as Lovasz and coauthors did is that the limit objects in the cut distance have a nice characterization: they're precisely graphons (i.e., symmetric, Lebesgue measurable functions from the unit square to the unit interval), which have a nice geometric characterization. Furthermore, there is the connection between cut distance and the convergence of subgraph frequencies. Do any such properties hold for the distance proposed in this paper? It seems insufficient that the authors just point out that one can define graph limits as limits of Cauchy sequences of graphs, because that can be done for absolutely any metric between graphs.

They do give some intuition regarding possible advantages of their distance over the cut distance, but I'm not

sure that they really follow through with this and prove theorems that make this concrete. For instance, can

they formalize their intuition to exhibit certain pairs of growing graph sequences that converge to the same

limit in cut distance but diverge with respect to their distance? Along the same lines, they should compare

with the convergence results in the Benjamini-Schramm framework and other notions of convergence for sparse

graph sequences. It might be that their distance metrizes the topology corresponding to one of these

other convergence notions.

4.) On page 16, they mention Theorem 7 in the past tense, but Theorem 7 appears much later in the paper.

6. PLOS authors have the option to publish the peer review history of their article (what does this mean?). If published, this will include your full peer review and any attached files.

Reviewer #1: No

Reviewer #2: No

---

## [Author Response · Author response to Decision Letter 0]

21 Jan 2021

Please see the attached PDF of responses to reviewer comments.

---

## [Decision Letter · Decision Letter 1]

23 Mar 2021

Graph diffusion distance: Properties and efficient computation

PONE-D-20-13130R1

Dear Dr. Scott,

We’re pleased to inform you that your manuscript has been judged scientifically suitable for publication and will be formally accepted for publication once it meets all outstanding technical requirements.

Kind regards,

Gabriele Oliva, Ph.D

Academic Editor

PLOS ONE

Additional Editor Comments (optional):

Only one review was obtained, so I carefully checked the paper myself. I agree with the reviewer and I am recommending acceptance.

Reviewers' comments:

Reviewer's Responses to Questions

**Comments to the Author**

1. If the authors have adequately addressed your comments raised in a previous round of review and you feel that this manuscript is now acceptable for publication, you may indicate that here to bypass the “Comments to the Author” section, enter your conflict of interest statement in the “Confidential to Editor” section, and submit your "Accept" recommendation.

Reviewer #1: All comments have been addressed

2. Is the manuscript technically sound, and do the data support the conclusions?

Reviewer #1: Yes

3. Has the statistical analysis been performed appropriately and rigorously? 

Reviewer #1: Yes

4. Have the authors made all data underlying the findings in their manuscript fully available?

Reviewer #1: Yes

5. Is the manuscript presented in an intelligible fashion and written in standard English?

Reviewer #1: Yes

6. Review Comments to the Author

Reviewer #1: To reiterate, I think this is strong work, and I commend the authors for making the changes that they did. I hope that the current number of figures and length of the piece is preserved as much as possible, as it has turned into quite a thorough contribution to the graph distance literature. Thanks for Figures 1, 2, and 5.

Regarding the new discussion of within-ensemble graph distances, I think it's fine to not include the figure from the response document, as the description in Section 5.3.5 is adequate. It is nevertheless interesting and I thank the authors for running those simulations.

As for the point about the D^2 notation, I approve of the change made below Equation 8.

Lastly, I went through the attached code--works well. A small note--and this is certainly an issue with my python env--but I had to swap out the `solve_dense` function from the lapsolver package with another from the https://github.com/cheind/py-lapsolver repo. Worked fine in the end, but just FYI. I ended up reproducing the figure for within-ensemble graph distances.

7. PLOS authors have the option to publish the peer review history of their article (what does this mean?). If published, this will include your full peer review and any attached files.

Reviewer #1: No

---

## [Editor Report · Acceptance letter]

12 Apr 2021

PONE-D-20-13130R1 

Graph diffusion distance: Properties and efficient computation  

Dear Dr. Scott:

I'm pleased to inform you that your manuscript has been deemed suitable for publication in PLOS ONE. Congratulations! Your manuscript is now with our production department. 

Kind regards, 

on behalf of

Dr. Gabriele Oliva 

Academic Editor

PLOS ONE